# STAIR: Addressing Stage Misalignment through Temporal-Aligned Preference Reinforcement Learning

**Yao Luan**[*1], **Ni Mu**[*1], **Yiqin Yang**[†2], **Bo Xu**[2], **Qing-Shan Jia**[†1]

[1] Beijing Key Laboratory of Embodied Intelligence Systems,
Department of Automation, Tsinghua University, Beijing, China
[2] The Key Laboratory of Cognition and Decision Intelligence for Complex Systems,
Institute of Automation, Chinese Academy of Sciences, Beijing, China
{luany23,mn23}@mails.tsinghua.edu.cn
yiqin.yang@ia.ac.cn, jiaqs@tsinghua.edu.cn

## Abstract

Preference-based reinforcement learning (PbRL) bypasses complex reward engineering by learning rewards directly from human preferences, enabling better alignment with human intentions. However, its effectiveness in multi-stage tasks, where agents sequentially perform sub-tasks (e.g., navigation, grasping), is limited by **stage misalignment**: Comparing segments from mismatched stages, such as movement versus manipulation, results in uninformative feedback, thus hindering policy learning. In this paper, we validate the stage misalignment issue through theoretical analysis and empirical experiments. To address this issue, we propose **ST**age-Al**I**gned **R**eward learning (STAIR), which first learns a stage approximation based on temporal distance, then prioritizes comparisons within the same stage. Temporal distance is learned via contrastive learning, which groups temporally close states into coherent stages, without predefined task knowledge, and adapts dynamically to policy changes. Extensive experiments demonstrate STAIR's superiority in multi-stage tasks and competitive performance in single-stage tasks. Furthermore, human studies show that stages approximated by STAIR are consistent with human cognition, confirming its effectiveness in mitigating stage misalignment. Code is available at https://github.com/iiiiiii11/STAIR.

## 1 Introduction

Reinforcement Learning (RL) has achieved significant progress in various applications, including robotics [17, 5], gaming [39, 34, 26], and autonomous systems [2, 8, 25]. Yet, the effectiveness of RL relies heavily on well-designed reward functions, which often require substantial manual effort and expert knowledge. To address this challenge, preference-based reinforcement learning (PbRL) has emerged as a promising alternative. This approach leverages human preferences among different agent behaviors as the reward signal, thereby alleviating the need for complex reward design.

However, many real-world sequential tasks exhibit multi-stage structures [41, 20], a factor often overlooked in existing research. For example, in a robotic "go-fetch-return" task, where an agent retrieves an object from a distance, the agent must ① navigate to the object, ② grasp it, and ③ transport it to a target location, as illustrated in Figure 1. Current PbRL methods [22, 32] face challenges of **stage misalignment** when collecting human preferences for these multi-stage tasks. This issue arises when behaviors from different stages, such as navigation and grasping, are presented

---

[*]Yao Luan and Ni Mu contributed equally.
[†]Correspondence to Yiqin Yang and Qing-Shan Jia.

39th Conference on Neural Information Processing Systems (NeurIPS 2025).

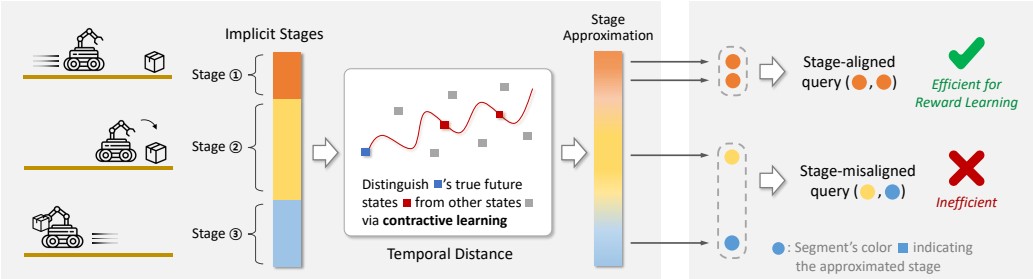

Figure 1: An overview of the proposed STAIR. (1) Learn the temporal distance to construct an on-policy stage difference approximator. (2) Use the temporal distance to select stage-aligned queries.

to humans for comparison. It leads to ambiguous feedback, as labelers struggle to compare behaviors in distinct subtasks, like efficient movement versus precise manipulation. This ambiguity can be clarified by event segmentation theory from cognitive science [45, 19]. The theory suggests that humans process action sequences by dividing them into discrete event boundaries. Consequently, comparisons involving behaviors that cross these natural event boundaries (from different stages) impose a higher cognitive load on labelers, thereby increasing ambiguity in their assessments. Moreover, significant differences in state distributions across stages can reduce the information gained from stage-misaligned queries, adversely impacting policy learning.

In this paper, we systematically analyze the impact of stage misalignment both theoretically and empirically. As illustrated in Proposition 2, comparing behaviors across different stages leads to inefficient policy learning, which requires significantly more feedback. Additionally, Proposition 3 and the experiments in Figure 3 reveal that when humans have inherent preferences for certain stages (e.g., favoring stages closer to task completion), conventional methods show quadratic growth in feedback demands, while the stage-aligned approach scales linearly. These findings underscore the importance of stage alignment for efficient preference learning in multi-stage tasks.

To address the stage misalignment issue, we focus on selecting queries with aligned stages for comparison, where a key challenge is identifying these stages without prior task knowledge. We propose a novel method, STAIR, which leverages temporal distance to measure stage differences. Temporal distance is learned through contrastive learning, grouping closely occurring states together, while separating those with significant temporal gaps. As illustrated in Figure 1, STAIR involves two main phases: 1) utilize contrastive learning to develop a temporal distance model for measuring stage differences; and 2) prioritize the comparison of segments with small stage differences. Extensive experiments show that STAIR outperforms state-of-the-art PbRL methods, achieving higher success rates, improved feedback efficiency, and faster convergence. Moreover, human studies validate that queries selected by STAIR align with human cognition, confirming its effectiveness in addressing stage misalignment.

In summary, our contributions are threefold: (1) We identify the critical issue of stage misalignment in PbRL through theoretical analysis and human experiments. (2) We propose STAIR, a novel stage-aligned learning method that automatically approximates stage differences via temporal distance, and selects stage-aligned queries to address the stage misalignment issue. (3) Extensive experiments show that STAIR outperforms existing methods, validating our key insight that stage alignment significantly improves preference learning efficiency in multi-stage scenarios.

## 2 Preliminaries

**Reinforcement learning.** An RL problem typically builds on a Markov Decision Problem (MDP) characterized by a tuple $(\mathcal{S}, \mathcal{A}, P, r, \mu_0, \gamma)$, where $\mathcal{S}$ and $\mathcal{A}$ are the state space and the action space, $P : \mathcal{S} \times \mathcal{A} \to \Delta(\mathcal{S})$ is the state transition, $r : \mathcal{S} \times \mathcal{A} \to \mathbb{R}$ is the reward function, $\mu_0$ is the initial state distribution, and $\gamma \in (0, 1)$ is the discount factor. A policy $\pi : \mathcal{S} \to \Delta(\mathcal{A})$ interacts with the environment by sampling action $a$ from the distribution $\pi(a|s)$ when observing state $s$. The goal of

RL agent is to learn a policy $\pi$, which maximizes the expectation of a discounted cumulative reward: $\mathcal{J}(\pi) = \mathbb{E}_{\mu_0, \pi} [\sum_{t=0}^{\infty} \gamma^t r(s_t, a_t)]$.

**Preference-based reinforcement learning.** In PbRL, the true reward function is not available and is replaced by an estimated reward function $\hat{r}_\psi$ parameterized by $\psi$, which is trained to be consistent with human preference. Specifically, given a segment pair $(\sigma_0, \sigma_1)$, where a segment $\sigma = \{s_k, a_k, \ldots, s_{k+H-1}, a_{k+H-1}\}$ is a state-action sequence with fixed length $H$, the preference $y$ is a binary indicating the index of the preferred segment. To construct $\hat{r}_\psi$, we follow the the Bradley-Terry model [4, 7], and define the preference predictor $P_\psi$ as follows:

$$P_\psi[\sigma_1 \succ \sigma_0] = \frac{\exp(\sum_{(s_t^1, a_t^1) \in \sigma_1} \hat{r}_\psi(s_t^1, a_t^1))}{\sum_{i \in \{0,1\}} \exp(\sum_{(s_t^i, a_t^i) \in \sigma_i} \hat{r}_\psi(s_t^i, a_t^i))}, \tag{1}$$

where $\sigma_1 \succ \sigma_0$ indicates the human prefer $\sigma_1$ than $\sigma_0$. Then, $\hat{r}_\psi$ can be trained by minimizing the cross-entropy loss between the true preference $y$ and the preference estimated by $P_\psi$:

$$\mathcal{L}_{\text{reward}}(\psi) = -\mathbb{E}_{(\sigma_0, \sigma_1, y) \sim \mathcal{D}^\sigma} \Big[ (1-y) \log P_\psi[\sigma_0 \succ \sigma_1] + y \log P_\psi[\sigma_1 \succ \sigma_0] \Big], \tag{2}$$

where $\mathcal{D}^\sigma$ is the preference buffer storing $(\sigma_0, \sigma_1, y)$ labeled by human.

**Stage Formulation.** For complex tasks with multi-stage characteristics, such as the "go-fetch-return" task in Figure 1, we reformalize the problem as a chain of stages as $\omega_1 \to \omega_2 \to \cdots \to \omega_{N_{\text{stage}}}$. Each stage $\omega$ represents an aggregation of states, and all stages comprise the stage space $\Omega = \{\omega_1, \ldots, \omega_{N_{\text{stage}}}\}$, $N_{\text{stage}} = |\Omega|$. A mapping function $F(s, \omega) : \mathcal{S} \times \Omega \to [0, 1]$ assigns the probability of a state $s$ belonging to stage $\omega$. Using $F(s, \omega)$, we can sequentially decomposed a trajectory $\tau = (s_0^\tau, a_0^\tau, \ldots, s_{T-1}^\tau, a_{T-1}^\tau)$ into $N_\tau$ segments $\{\sigma_i^\tau\}_{i=1}^{N_\tau}$ ($N_\tau \leq N_{\text{stage}}$), where all states within a segment belong to the same stage. Specifically, let $G^\tau(i)$ denote the index of the stage for state $s_i^\tau$, then, the segmentation of $\tau$ is obtained by solving the following optimization problem:

$$\max_{G^\tau(\cdot)} \mathbb{E}_{s_i^\tau \in \tau}[F(s_i^\tau, \omega_{G^\tau(i)})], \tag{3}$$
$$\text{s.t. } G^\tau(0) = 1, \ G^\tau(i) - 1 \leq G^\tau(i-1), \ G^\tau(i-1) \leq G^\tau(i), \ i \in \{1, \ldots, T-1\}.$$

This optimization problem maximizes the likelihood of the stage decomposition, quantified by $F$, while maintaining the sequential chain structure of stage transitions. Once $G^\tau(\cdot)$ is solved, segments can be derived as $\sigma_i^\tau = (s_j, a_j)_{G^\tau(j)=i}, i = 1, \ldots, N_\tau$, where $N_\tau = \max_i G^\tau(i)$ is the number of stages in trajectory $\tau$. $N_\tau = N_{\text{stage}}$ holds only if the task is completed within this trajectory. The optimal objective value of problem (3) quantifies the degree to which the trajectory $\tau$ can be divided into stages. To evaluate the degree to which an MDP can be divided into stages, we propose using the average optimal objective value of problem (3) across trajectories generated by the optimal policy $\pi^*$. This measure can be formally expressed as: $\mathcal{F} \triangleq \mathbb{E}_{\tau \sim \pi^*}[\max_{G^\tau(\cdot)} \mathbb{E}_{s_i^\tau \in \tau}[F(s_i^\tau, \omega_{G^\tau(i)})]]$.

## 3 Impact of Stage Misalignment

### 3.1 Impact of Stage Misalignment in Multi-Stage Tasks

In this subsection, we first validate the existence of the multi-stage property in real-world tasks, then demonstrate the negative effects of stage misalignment. To verify the multi-stage property, we analyze the state distributions at each timestep, as greater differences among them suggest a higher potential for problem segmentation into stages. Formally, we introduce Proposition 1 (proof in Appendix A), which considers a classifier that predicts the collected timestep for a given state, with high accuracy suggesting the task's multi-stage property. Note that Proposition 1 holds independently of $N_{\text{stage}}$.

**Proposition 1.** *For an MDP and trajectories generated by its optimal policy $\pi^*$, consider a calibrated classifier $\hat{T}(s)$ that takes a state $s$ as input and outputs the probability $p_t(s)$ that the state $s$ is collected at step $t \in \{1, 2, \ldots N\}$, $\hat{T}(s) = \max_t p_t(s)$. Denote the accuracy of the classifier as acc, the multi-stage measure $\mathcal{F}$ has a lower bound $\mathcal{F} \geq acc^2$.*

To empirically verify the multi-stage property in real-world scenarios, we analyze MetaWorld robotic manipulation tasks [44] as an example, since most MetaWorld tasks are multi-stage. For example,

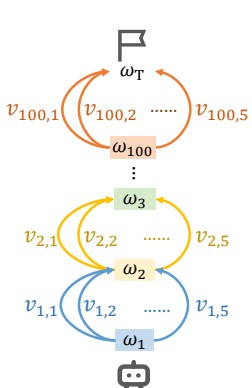

Figure 2: An illustration of the experiment in the abstract MDP model.

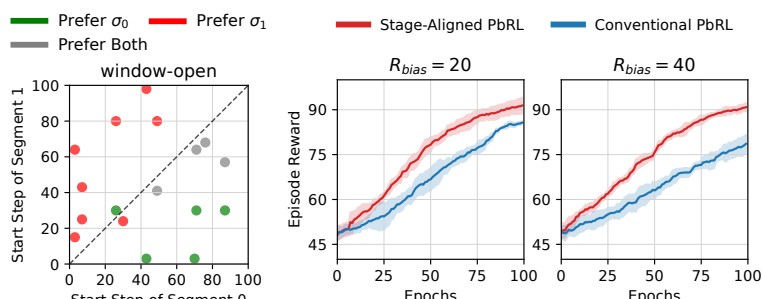

Figure 3: **(Left)** Human preferences of segments started at different timesteps in the window-open task. Each point $(t_x, t_y)$ represents that segment $\sigma_0$ and $\sigma_1$ are collected from steps $t_x$ and $t_y$. Humans prefer segments in later timesteps, suggesting a stage reward bias, where humans' underlying reward is higher in these later stages. **(Right)** Normalized episode reward of various $R_{\text{bias}}$ in the abstract MDP model. Additional results are shown in Appendix F.1.

the window-open task requires the arm to open the window, which can be divided into two stages: grasping the handle and pulling the handle. Specifically, we train a classifier as in Proposition 1 using trajectories of the expert policy. The classifier achieves an impressive 81% accuracy, indicating significant differences in state distributions across timesteps, thereby supporting the presence of the multi-stage property in these tasks. Experimental details are shown in Appendix D.1.

Then, we theoretically evaluate the impact of stage misalignment on efficiency in multi-stage tasks. We reformalize these tasks to an abstract MDP $(\Omega, \{\Upsilon_i\}_{i=1}^{|\Omega|}, \bar{P}, \bar{r}, \bar{\mu}_0, \bar{\gamma})$, where each state $w_i$ represents a stage[1] and has $|\Upsilon_i|$ actions $\{\upsilon_{i,j}\}_{j=1}^{|\Upsilon_i|}$. Transitions follow a deterministic chain structure ($\omega_i \to \omega_{i+1}$), and the reward function $\bar{r}$ perfectly aligns with human preferences. With this formulation, we compare two query selection methods: (1) *stage-aligned selection* samples segments within the same stage, and (2) *conventional selection* samples segments across all stages [7, 22]. As analyzed by Proposition 2, generally, the conventional PbRL requires more queries to learn an optimal policy than the stage-aligned PbRL. The analysis leverages the similarity between PbRL and ranking methods, where the true ordering is defined by the reward function $\bar{r}(\omega, \upsilon)$. The stage-aligned method learns local orderings for actions in each stage, and the conventional method learns a global ordering of all stage-action pairs. Detailed assumptions and the proof are provided in Appendix A.

**Proposition 2.** *In the worst case scenario, the conventional PbRL needs $\mathcal{O}(|\Omega||\Upsilon| \log(|\Omega||\Upsilon|))$ additional queries to learn the optimal policy compared to the stage-aligned PbRL.*

### 3.2 Impact of Stage Misalignment Under Stage Reward Bias

While Section 3.1 analyzes the impact of stage misalignment in general multi-stage tasks, this subsection focuses on a specific case, where humans prefer certain stages over others. Our theoretical and empirical analysis shows that stage misalignment has a more pronounced effect in these scenarios.

Assuming human preferences can be modeled by a reward function, we formalize these multi-stage tasks through a reward decomposition: $r(s, a) = r_{\text{stage}}(s) + r_{\text{sa}}(s, a)$, where $r_{\text{stage}}(s)$ represents the average reward of the stage containing state $s$, and $r_{\text{sa}}(s, a)$ denotes the reward advantage of the state-action pair within that stage. This decomposition reflects that stages differ in importance in human preference judgments. We refer to $r_{\text{stage}}(s)$ as the *stage reward bias*, since a higher stage reward bias reflects more preferred stages. Note that this decomposition is primarily for validating the severity of stage misalignment, and is not necessary for the proposed method (Section 4).

To validate the existence of stage reward bias in practical tasks, we conduct a human preference experiment. We sample segments of 20-timestep lengths from expert trajectories, and collect human preference labels for segment pairs. Human labelers watch video renderings of each segment and

---

[1]To avoid confusion with the general MDP formulation, we use $\omega$ to denote both the state and the stage for the abstract MDP, and use $\upsilon$ to denote the action.

select the one that is more beneficial for achieving the task objective, as detailed in Appendix D.1. Results in Figure 3 (Left) indicate a clear human preference for segments starting later, which empirically demonstrates the existence of stage reward bias in human preference, where certain stages receive higher valuation than others.

Further, we evaluate the benefit of stage-aligned query selection under stage reward bias. Specifically, we instantiate the abstract MDP model as in Figure 2, where $\omega_i, \upsilon_{i,j}$ denote the $i$-th stage and the $j$-th action in stage $\omega_i$. In this model, $|\Omega| = 101, |\Upsilon_i| = 5, i \in \{1, \cdots, 100, \mathrm{T}\}$ ($w_\mathrm{T}$ is a terminal state) and $\bar{r}_\mathrm{stage}(\omega_i) \sim \mathrm{Uniform}[0, R_\mathrm{bias}], \bar{r}_\mathrm{sa}(\omega_i, \upsilon_j) \sim \mathrm{Uniform}[0, 10]$, with $R_\mathrm{bias}$ controls the scale of stage reward bias. As shown in Figure 3 (Right), stage-aligned selection yields better learning efficiency than conventional methods in this model. This advantage may result from stage-misaligned queries providing limited information for action selection in the current stage, especially under significant stage reward bias. Experimental details are provided in Appendix D.1.

We also provide a theoretical analysis on query complexity to support these findings. Proposition 3 analyzes cases with significant stage reward bias, yielding a more precise result than Proposition 2. Detailed assumptions and proofs are available in Appendix A.

**Proposition 3.** *If the stage reward bias is sufficiently large, such that the reward ordering between $(\omega_i, \upsilon_j)$ and $(\omega_{i'}, \upsilon_{j'})$ depends solely on $\omega_i$ and $\omega_{i'}$ for $i \neq i'$, then in the worst case scenario, conventional PbRL requires $\mathcal{O}(|\Omega|^2|\Upsilon|\log(|\Upsilon|))$ additional queries to learn the optimal policy compared to stage-aligned PbRL.*

In practice, the stage reward bias is often less significant than that assumed in Proposition 3. Consequently, the sample complexity for conventional PbRL ranges from $\mathcal{O}(|\Omega||\Upsilon|\log(|\Omega||\Upsilon|))$ and $\mathcal{O}(|\Omega|^2||\Upsilon|\log(|\Upsilon|))$, which remains considerably higher than that of stage-aligned PbRL. In summary, in multi-stage tasks, stage-aligned query selection enhances policy learning by eliminating less informative queries, especially when stage-specific bias is significant.

## 4 Stage-Aligned Reward Learning

To address the stage misalignment in PbRL, we focus on selecting stage-aligned queries. However, two key challenges arise: First, stage definitions are often based on subjective expertise of specific tasks, limiting their generalizability across different tasks. Second, stage measurement should be adaptable to evolving policies; otherwise, stages assessed with early suboptimal policies may become incompatible with the newer, improved ones. We elaborate on these challenges in Appendix C.

To tackle these challenges, we propose **ST**age-A**lI**gned **R**eward learning (STAIR). A core innovation of STAIR is the use of temporal distance to develop an on-policy stage difference approximator. First, STAIR employs contrastive learning to construct a temporal distance model, which efficiently approximates stage differences between states in an on-policy manner, as detailed in Section 4.1. Then, we design a query selection method in Section 4.2 to identify stage-aligned queries. The overall framework of STAIR is illustrated in Figure 1 and Algorithm 1.

### 4.1 Learning Temporal Distance

In STAIR, we utilize temporal distance as a measure of stage differences, as it effectively addresses the aforementioned challenges. Specifically, temporal distance quantifies the transition probabilities between states under a given policy. Easily reachable state pairs have smaller temporal distances, indicating similar stages, whereas hard-to-reach state pairs show larger distances, suggesting distinct stages. This measure does not rely on any predefined task-specific stage definitions, and could adapt dynamically to policy changes, as temporal distance can be learned in an on-policy manner. Therefore, temporal distance serves as an effective stage approximator.

Let $p_\gamma^\pi(s_+ = y|s_0 = x)$ denote the discounted state occupancy measure of state $y$ when starting from $x$ under policy $\pi$. It represents the discounted probability of reaching state $y$ from $x$:

$$p_\gamma^\pi(s_+ = y|s_0 = x) = (1 - \gamma) \sum_{k=0}^{\infty} \gamma^k p^\pi(s_k = y|s_0 = x), \tag{4}$$

where $p^\pi(s_k = y|s_0 = x)$ is the probability of reaching $y$ from $x$ in exactly $k$ steps under policy $\pi$. Using the discounted state occupancy measure in (4), temporal distance quantifies the transition

probability between states. The successor distance [30] serves as a start-of-the-art implementation of the temporal distance, which is defined as

$$d_{\text{SD}}^{\pi}(x, y) = \log\left(p_{\gamma}^{\pi}(s_+ = y | s_0 = y) / p_{\gamma}^{\pi}(s_+ = y | s_0 = x)\right). \quad (5)$$

The successor distance is proven to be a quasimetric [30], even in stochastic MDPs, making it a reliable measure of the stage difference between states.

To learn the successor distance $d_{\text{SD}}^{\pi}(x, y)$, we employ contrastive learning [30]. This method trains an energy function $f_{\theta}(x, y)$ that assigns higher scores to state pairs $(x, y)$ belonging to the same trajectory, which follows the joint distribution $p_{\gamma}^{\pi}(s_+ = y | s_0 = x)$; and that assigns lower scores to pairs $(x, y)$ from different trajectories, which are equivalently sampled from the marginal distributions $p_s(x)$ and $p_+(y) \triangleq \int_s p_s(x) p_{\gamma}^{\pi}(s_f = y | s_0 = x)$ respectively. The energy function $f_{\theta}(x, y)$ is optimized using the symmetrized infoNCE loss [31]:

$$\mathcal{L}_{\theta} = \sum_{i=1}^{B}\left[\log\frac{\exp(f_{\theta}(x_i, y_i))}{\sum_{j=1}^{B}\exp(f_{\theta}(x_i, y_j))} + \log\frac{\exp(f_{\theta}(x_i, y_i))}{\sum_{j=1}^{B}\exp(f_{\theta}(x_j, y_i))}\right]. \quad (6)$$

As the optimal energy function $f_{\theta^*}$ satisfies $f_{\theta^*}(x, y) = \log\left(\frac{p_{\gamma}^{\pi}(s_+ = y | s_0 = x)}{C \cdot p_+(y)}\right)$ [35], the successor distance could be derived by $d_{\text{SD}}^{\pi}(x, y) = f_{\theta^*}(y, y) - f_{\theta^*}(x, y)$. Following prior work [30], we parameterize the score function as $f_{\theta}(x, y) = c_{\theta_c}(y) - d_{\theta_d}(x, y)$, where $\theta = (\theta_c, \theta_d)$. Then after contrastive learning, we have $d_{\text{SD}}^{\pi}(x, y) = d_{\theta_d^*}(x, y)$, allowing $d_{\theta_d^*}(x, y)$ to serve directly as the temporal distance.

## 4.2 Stage-Aligned Query Selection

In this subsection, we focus on addressing the stage misalignment issue via temporal distance. Specifically, we propose a stage-aligned query selection method, which mitigates stage misalignment by filtering out queries identified as misaligned based on temporal distance.

**Measuring stage difference between segments.** The query selection method requires measuring the stage difference between two segments $\sigma_0, \sigma_1$. However, the temporal distance in Section 4.1 was originally designed to measure state distances. Therefore, we propose a quadrilateral distance to adapt the temporal distance in Section 4.1 for segment distances:

$$d_{\text{stage}}(\sigma_0, \sigma_1) = \frac{1}{4} \cdot (d_{\text{SD}}^{\pi}(s_0^0, s_0^1) + d_{\text{SD}}^{\pi}(s_{H-1}^0, s_{H-1}^1) + d_{\text{SD}}^{\pi}(s_0^0, s_{H-1}^1) + d_{\text{SD}}^{\pi}(s_{H-1}^0, s_0^1)), \quad (7)$$

where $s_0^i$ and $s_{H-1}^i$ denote the initial and final states of segment $\sigma_i$. This quadrilateral distance ensures that stage-aligned segments yield smaller $d_{\text{stage}}$ values. As visualized in Figure 4, the side terms, $d_{\text{SD}}^{\pi}(s_0^0, s_0^1) + d_{\text{SD}}^{\pi}(s_{H-1}^0, s_{H-1}^1)$, favor segments with closely aligned starting and ending points, indicating straightforward stage alignment. The diagonal terms, $d_{\text{SD}}^{\pi}(s_0^0, s_{H-1}^1) + d_{\text{SD}}^{\pi}(s_{H-1}^0, s_0^1)$, favor segments with shorter temporal spans, thereby focusing on segments concentrated within a single stage. We further analyze the behaviour of $d_{\text{stage}}$ theoretically in Appendix A.3.

Figure 4: An illustration of the quadrilateral distance defined in (7). The distance $d_{\text{stage}}(\sigma_0, \sigma_1)$ is the average length (measured by temporal distance $d_{\text{SD}}^{\pi}$) of the four dashed lines in the quadrilateral.

**Query selection.** Using the stage difference metric, we then propose the stage-aligned query selection method in Algorithm 2. This method calculates a score function $I(\sigma_0, \sigma_1)$ for each candidate segment pair $(\sigma_0, \sigma_1)$, and selects queries with the largest scores. To enhance the informativeness of comparisons, in the score function, we integrate stage alignment with the reward model's uncertainty, a widely adopted metric of informativeness [22, 38]. We represent this uncertainty with $d_{\text{state}}(\sigma_0, \sigma_1)$, which calculates the variance of $P_{\psi}[\sigma_0 \succ \sigma_1]$ value across ensemble members: $d_{\text{state}}(\sigma_0, \sigma_1) = \text{Var}[P_{\psi_i}[\sigma_0 \succ \sigma_1]_{i=1}^{n_e}]$, following [22]. Here $P_{\psi_i}[\sigma_0 \succ \sigma_1]_{i=1}^{n_e}$ denotes an ensemble of $n_e$ identical reward models with different randomly initalized parameters. Formally, the selection score is defined as:

$$I(\sigma_0, \sigma_1) = (c_{\text{stage}} - d_{\text{stage}}(\sigma_0, \sigma_1))(c_{\text{state}} + d_{\text{state}}(\sigma_0, \sigma_1)), \quad (8)$$

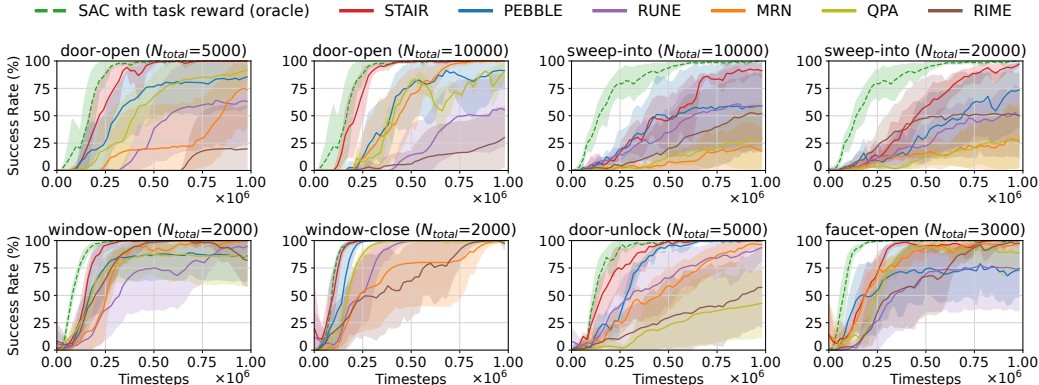

Figure 5: Learning curves on robot manipulation tasks from MetaWorld. The solid line and the shaded area represent the mean and the standard deviation of success rates (%).

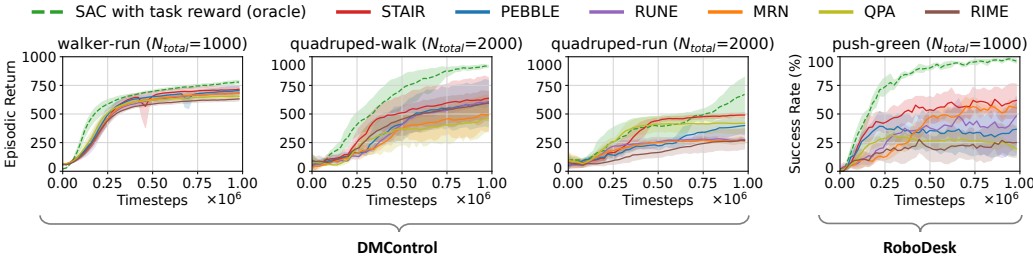

Figure 6: Learning curves on locomotion tasks from DMControl and a robot manipulation task from RoboDesk. The solid line and the shaded area represent the mean and the standard deviation of the episodic returns (DMControl) and success rates (RoboDesk).

where $d_{\text{stage}}$ and $d_{\text{state}}$ are normalized to $[0, 1]$, and $c_{\text{stage}}, c_{\text{state}}$ are hyperparameters. The first term encourages stage alignment, while the second term emphasizes queries where the reward model shows high uncertainty.

## 5 Experiments

We design our experiments to answer the following questions: *Q1*: How does STAIR compare to other state-of-the-art methods in multi-stage tasks? *Q2*: Is the stage-aligned query selection still beneficial in single-stage tasks? *Q3*: Does the stage approximated by temporal distance align with human cognition? *Q4*: What is the contribution of each of the proposed techniques in STAIR?

### 5.1 Setup

**Domains.** We evaluate STAIR on several complex robotic manipulation and locomotion tasks from MetaWorld [44], DMControl [40], RoboDesk [16]. In RoboDesk, we modify the task representation from a pixel-based camera image to a robot-arm state representation, which aligns with MetaWorld. Details of these domains are shown in Appendix D.2. MetaWorld and RoboDesk focus on multi-stage tasks that achieve specific objectives, such as opening a window, which requires an arm to first grasp the handle and then pull it. Conversely, DMControl includes single-stage tasks, which focus on maximizing travel distance or velocity, posing challenges in defining stages for humans. We evaluate STAIR in both multi-stage and single-stage domains to show its robustness and generalizability.

**Baselines and Implementation.** We compare STAIR with several state-of-the-art PbRL methods, including PEBBLE [22], RUNE [23], MRN [24], RIME [6] and QPA [13]. We also evaluate SAC [11] with ground truth reward as a performance upper bound. For PEBBLE and RUNE, we employ disagreement-based query selection, as it yields the best performance. Following prior works

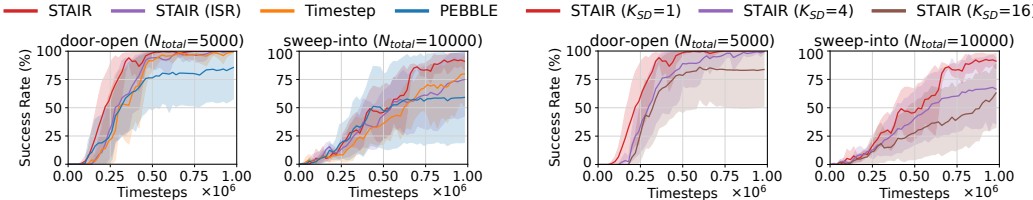

Figure 7: Ablation results. **(Left)** Performance of STAIR with various stage difference approximators. **(Right)** Performance of STAIR under a range of temporal distance update frequencies $K_{\text{SD}}$.

[21, 22, 23], we consider an oracle script teacher that provides preference by comparing the total task reward of two segments. Please refer to Appendix D.3 for further details.

## 5.2 Results on Benchmark Tasks

**Multi-stage tasks in MetaWorld and RoboDesk.**  As shown in Figure 5 (MetaWorld) and 6 (RoboDesk), STAIR outperforms baselines across all evaluated multi-stage tasks, achieving success rates close to 100% in most cases. Moreover, STAIR shows a faster convergence speed. For example, in door-open ($N_{\text{total}}$=5000) and window-open ($N_{\text{total}}$=2000), STAIR's performance rapidly improves over fewer time steps, and reaches a stable high performance earlier than the other baselines.

**Single-stage tasks in DMControl.**  As shown in Figure 6, STAIR outperforms PEBBLE and is competitive with other baselines, even in single-stage tasks. This suggests the potential of STAIR for broader applications. This success may arise from the implicit curriculum learning induced by STAIR: Though the task is not multi-stage, STAIR implicitly divides the reward learning process into stages by introducing queries progressively. Later learning stages are presented only after the agent masters the earlier ones, enabling the model to focus on the complexities of the newly added stages. We provide further discussions on it in Appendix G, and will explore this further in future work.

## 5.3 Human Experiments

We conducted a human labeling experiment to examine whether the stages approximated by temporal distance align with human cognition. Specifically, human labelers are instructed to review queries generated by STAIR and PEBBLE, and assess whether the two segments in a query correspond to the same stage, as detailed in Appendix E. Figure 8 shows the ratio of queries identified by labelers as stage-aligned. Additionally, Figure 13 visualizes segment pairs selected by STAIR and PEBBLE, where the pair selected by STAIR shows different behaviors in similar stages, and the one of PEBBLE exhibits different stages. The results indicate that STAIR effectively selects queries recognized by humans as stage-aligned, thereby facilitating human labeling.

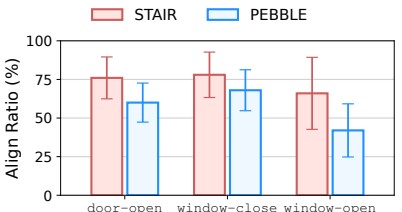

Figure 8: The ratio of two selected segments in a query being at the same stage from a human perspective.

## 5.4 Ablation Study

**Impact of the stage difference approximator.**  To show the efficiency of the segment stage difference approximator in (7), we design two variants of STAIR: (1) **Timestep**: This variant uses the collected timestep of a state as the stage approximation, then uses *Interval Span Ratio* (ISR) $d_{\text{ISR}}(\sigma_0, \sigma_1) = \frac{\tilde{\omega}_{0,\max} - \tilde{\omega}_{1,\min}}{\tilde{\omega}_{1,\max} - \tilde{\omega}_{0,\min}}$ to assess the stage difference between two segments $\sigma_0, \sigma_1$, where $\tilde{\omega}_{i,\min}$ and $\tilde{\omega}_{i,\max}$ are the minimum and maximum approximated stage for states in $\sigma_i$. It is assumed that $\tilde{\omega}_{0\,\min} \leq \tilde{\omega}_{1,\min}$, otherwise, we swap $\sigma_0$ and $\sigma_1$. The ISR quantifies the alignment between two intervals by measuring their intersection if they overlap, or the gap between them otherwise, thereby serving as an indicator of stage alignment. (2) **STAIR (ISR)**: This variant uses the temporal distance from the current state to the trajectory's initial state as the stage approximation, which serves as

Table 1: Performance of STAIR with various $c_{\text{stage}}$ and $c_{\text{state}}$ values.

| $c_{\text{stage}}$ | $c_{\text{state}}$ | door-open ($N_{\text{total}} = 5000$) | sweep-into ($N_{\text{total}} = 10000$) |
|---|---|---|---|
| 1 | 0.1 | $99.91 \pm 0.07$ | $87.81 \pm 7.57$ |
| 1 | 0.5 | $99.53 \pm 0.44$ | $92.31 \pm 2.28$ |
| 2 | 0.1 | $100.00 \pm 0.00$ | $91.08 \pm 3.42$ |
| 2 | 0.5 | $98.67 \pm 1.14$ | $93.58 \pm 4.82$ |

Table 2: Performance of STAIR and PEBBLE on door-open using different numbers of query feedback.

| $N_{\text{total}}$ | STAIR | PEBBLE |
|---|---|---|
| 500 | $52.01 \pm 23.18$ | $20.00 \pm 17.88$ |
| 2000 | $77.77 \pm 11.67$ | $28.79 \pm 17.02$ |
| 5000 | $100.00 \pm 0.00$ | $85.57 \pm 12.77$ |
| 10000 | $99.93 \pm 0.06$ | $92.53 \pm 6.53$ |

the on-policy version of timestep, and also uses ISR to evaluate the stage difference between two segments.

As shown in Figure 7 (Left), STAIR outperforms all the variants. The limitation of these variants may lie in measuring stage differences between segments in a one-dimensional axis. In contrast, our quadrilateral distance evaluates stage differences in a two-dimensional space, effectively modeling more complex relationships between segments.

**Impact of temporal distance update frequency.** Accurately estimating stage differences requires the stage approximator to adapt effectively to the evolving policy. Figure 7 (Right) shows that a lower frequency of temporal distance updates (larger $K_{\text{SD}}$) leads to a decreased performance of STAIR. This emphasizes the importance of training the stage difference approximator in an on-policy manner.

**Robustness on query selection hyperparameters.** Table 1 shows that STAIR's performance remains consistent across hyperparameter configurations. This shows the robustness of STAIR to hyperparameter changes, which ensures stable and effective learning in more general settings.

**Enhanced feedback efficiency.** We compare the performance of STAIR and PEBBLE using different numbers of queries ($N_{\text{total}}$) on the MetaWorld door-open task. Table 2 shows that STAIR consistently outperforms PEBBLE, demonstrating its effectiveness in utilizing limited feedback. This result arises from STAIR's stage-aligned query selection, which offers more informative queries for policy learning, enabling the agent to learn effectively with fewer feedback.

## 6 Related Work

**Preference-based reinforcement learning.** PbRL has emerged as a promising framework for aligning agent behaviors with human intentions, thereby alleviating the need for complex reward engineering [7, 21, 27, 28]. To enhance feedback efficiency, prior works have explored unsupervised pre-training [22], semi-supervised data augmentation [32, 1], reward uncertainty-based exploration [23], on-policy query selection [13], and meta-learning [12]. These methods aim to minimize dependence on human feedback by selecting more informative queries based on entropy [14], ensemble disagreement [22], or feature-space diversity [3]. However, they often overlook the inherent multi-stage nature of real-world tasks [41, 20], where comparisons of cross-stage segments can lead to inefficient learning. Our approach addresses this by introducing stage-aligned query selection based on temporal distance, which improves learning efficiency by focusing on segment comparisons within the same stage.

**Temporal distance learning in RL.** Temporal distance quantifies the expected transition steps between states under a given policy, which is essential in goal-conditioned RL [9], skill discovery [33], and state representation learning [37]. Prior works learn it through spectral decomposition of state transitions [43], constrained optimization for temporal consistency [42], and contrastive learning that groups temporally adjacent states [30]. Recent research [30] introduced successor distance, which is formally guaranteed to be a quasimetric. The quasimetric characteristic ensures the successor distance to be a reliable state similarity measure, which thus has been employed to design intrinsic rewards that enhance exploration [15]. Nevertheless, these methods remain underexplored in multi-stage sequential decision problems. Our work fills this gap by employing temporal distance as a stage similarity measure, enabling stage decomposition without task knowledge.

# 7 Conclusion

This paper presents STAIR, a novel approach for addressing stage misalignment in multi-stage environments. STAIR first constructs a stage difference approximator via temporal distance, learned through contrastive learning. Then, STAIR extends the temporal distance to assess segment distances via quadrilateral distance, enabling the selection of stage-aligned queries. Experiments show that STAIR outperforms baselines in multi-stage tasks and remains competitive in single-stage domains. Human experiments further confirm the effectiveness of the learned stage approximation.

**Limitations.** One limitation of STAIR is that the quadrilateral distance only assesses pairwise segment differences, limiting its applicability to other preference formats. We will explore this in future work.

## Acknowledgments and Disclosure of Funding

This work is supported by NSFC (No. 62125304), the National Key Research and Development Program of China (2022YFA1004600), the 111 International Collaboration Project (BP2018006), the Beijing Natural Science Foundation (L233005), and BNRist project (BNR2024TD03003).

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

# A Proof

## A.1 Proof of Proposition 1

**Proposition 4.** *For an MDP and trajectories generated by its optimal policy $\pi^*$, consider a classifier $\hat{T}(s)$ that takes a state $s$ as input and outputs the probability $p_t(s)$ that the state $s$ is collected at step $t \in \{1, 2, \ldots N\}$, $\hat{T}(s) = \max_t p_t(s)$. Denote the accuracy of the classifier as $acc$, the multi-stage measure $\mathcal{F}$ is lower bounded by $acc \cdot \mathbb{E}_{\tau \sim \pi^*, s \in \tau} [\max_t p_t(s)]$.*

*Proof.* We prove this proposition by constructing a stage chain with the classifier.

First, we aggregate the timesteps into $|\Omega|$ sets by defining

$$t_i^+ = \{t | \left\lceil \frac{t}{T/|\Omega|} \right\rceil = i\}. \tag{9}$$

Then, we define an aggregated classifier $\hat{T}^+(s)$ that receives a state and outputs the probability $p_{t_i^+}(s)$ that the given state is collected at the aggregated step $t_i^+, i \in \{1, 2, \ldots |\Omega|\}$, Specifically, we have

$$p_{t_i^+}^+(s) = \sum_{t \in t_i^+} p_t(s), \tag{10}$$

and the aggregated classifier is defined as

$$\hat{T}^+(s) = \max_{t_i^+} p_{t_i^+}^+(s). \tag{11}$$

The accuracy of the classifier is given by

$$acc = \mathbb{E}_{\tau \sim \pi^*, s \in \tau} [\mathbb{I}(t^*(s, \tau) = \arg \max_t p_t(s))], \tag{12}$$

where $t^*(s, \tau)$ is the step of state $s$ in trajectory $\tau$. Therefore, the accuracy of the aggregated classifier is lower bounded by $acc$, since any errors made by the aggregated classifier must also be present in the original classifier, while the reverse is not true.

Note that the aggregated timestep set shows a chain structure. We divide the stages according to these aggregated timestep sets, where the stage $\omega_i$ corresponds to the aggregated timestep set $t_i^+$. Then we model the probability of state $s$ being in stage $\omega_i$ using the aggregated classifier:

$$F(s, \omega_i) = p_{t_i^+}(s). \tag{13}$$

If the classifier is perfectly accurate, i.e. $acc = 1$, the chain constraints defined in (3) are naturally satisfied since the stages are determined by the aggregated time steps. In this ideal scenario, we achieve a multi-stage measure represented as:

$$\mathcal{F}^* = \mathbb{E}_{\tau \sim \pi^*, s \in \tau} [\max_i p_{t_i^+}^+(s)]. \tag{14}$$

However, in practice, $acc \neq 1$. In this case, each state $s$ has a worst-case probability $1 - acc$ of being assigned to a wrong stage $\omega^-$, where $\omega^- \neq \omega_{\lceil \frac{t^*(s,\tau)}{T/|\Omega|} \rceil}$. Consequently, the mapping probability $F$ of the true stage is

$$F(s, \omega_{\lceil \frac{t^*(s,\tau)}{T/|\Omega|} \rceil} | \omega_{\lceil \frac{t^*(s,\tau)}{T/|\Omega|} \rceil} \neq \arg \max_{\omega_j} F(s, \omega_j)) \geq 0. \tag{15}$$

Therefore, we could derive a lower bound of $\mathcal{F}$:

$$\mathcal{F} \geq acc \cdot \mathbb{E}_{\tau \sim \pi^*, s \in \tau} \left[ \max_i p_{t_i^+}^+(s) \right] = acc \cdot \mathbb{E}_{\tau \sim \pi^*, s \in \tau} \left[ \max_i \sum_{t \in t_i^+} p_t(s) \right]$$

$$\geq acc \cdot \mathbb{E}_{\tau \sim \pi^*, s \in \tau} \left[ \max_t p_t(s) \right], \tag{16}$$

which concludes the proof. $\square$

**Lemma 1.** *For a multi-class classification problem, where $\mathcal{X}$ is the input space, $\mathcal{Y} = \{1, 2, \ldots, K\}$ is the set of classes. Consider a classifier $f : \mathcal{X} \to \Delta\mathcal{Y}$, and let $f_k(x)$ denote the predicted probability that sample $x$ belongs to class $k$. Suppose the data $(x, y(x))$ (or $(x, y)$ for simplicity) follows the joint distribution $\mathcal{P}$. Define the accuracy of the classifier as $acc = \mathbb{E}_{(x,y)\sim\mathcal{P}}[\mathbb{I}(y = \arg\max_k f_k(x))]$. If the classifier is perfectly calibrated, i.e., for all classes $k \in \mathcal{Y}$ and probability $p \in [0, 1]$, we have $\Pr(y = k | f_k(x) = p) = p$, then we have $acc = \mathbb{E}_{x\sim\mathcal{P}}[\max_k f_k(x)]$.*

*Proof.* Since the classifier is perfectly calibrated, we have

$$
\begin{aligned}
& \Pr(y(x) = k | f_k(x) = p) = p \\
\Leftrightarrow\quad & \frac{\Pr(y(x) = k, f_k(x) = p)}{\Pr(f_k(x) = p)} = p \\
\Leftrightarrow\quad & \frac{\Pr(x)\Pr(y(x) = k, f_k(x) = p | x)}{\Pr(x)\Pr(f_k(x) = p | x)} = p \\
\Leftrightarrow\quad & \frac{\Pr(x)\Pr(y(x) = k | x)\Pr(f_k(x) = p | x)}{\Pr(x)\Pr(f_k(x) = p | x)} = p \\
\Leftrightarrow\quad & \Pr(y(x) = k | x) = p = f_k(x),
\end{aligned}
\tag{17}
$$

where $\Pr(x)$ represents the marginal distribution of sample $x$. The fourth line is because the classifier is conditionally independent of the ground truth label given the sample $x$.

Then, we focus on the accuracy:

$$
\begin{aligned}
acc &= \mathbb{E}_{(x,y)\sim\mathcal{P}}[\mathbb{I}(y = \arg\max_k f_k(x))] \\
&= \mathbb{E}_x \sum_y \Pr(y|x)[\mathbb{I}(y = \arg\max_k f_k(x))] \\
&= \mathbb{E}_x \sum_y f_y(x)[\mathbb{I}(y = \arg\max_k f_k(x))] \\
&= \mathbb{E}_x[f_{\arg\max_k f_k(x)}(x)] \\
&= \mathbb{E}_x[\max_k f_k(x)],
\end{aligned}
\tag{18}
$$

where the third equation is obtained by substituting (17). That concludes the proof. $\square$

**Proposition 1.** *For an MDP and trajectories generated by its optimal policy $\pi^*$, consider a calibrated classifier $\hat{T}(s)$ that takes a state $s$ as input and outputs the probability $p_t(s)$ that the state $s$ is collected at step $t \in \{1, 2, \ldots N\}$, $\hat{T}(s) = \max_t p_t(s)$. Denote the accuracy of the classifier as $acc$, the multi-stage measure $\mathcal{F}$ has a lower bound $\mathcal{F} \geq acc^2$.*

*Proof.* Using Proposition 4, we have

$$
\mathcal{F} \geq acc \cdot \mathbb{E}_{\tau\sim\pi^*, s\in\tau}\left[\max_t p_t(s)\right],
\tag{19}
$$

Using Lemma 1, we have $\mathbb{E}_{\tau\sim\pi^*, s\in\tau}[\max_t p_t(s)] = acc$, which leads to the conclusion straightforwardly. $\square$

## A.2   Proof of Proposition 2 and 3

The proofs are based on the abstract MDP introduced in Section 3.1. This MDP is fully stage-wise, with a discrete state space $\Omega$ and a discrete action space $\Upsilon$, where each state comprises a stage. In addition, we make the following assumptions:

- Human preferences are perfectly aligned with an oracle reward function $\bar{r}(\omega, \upsilon)$. The estimated reward function $\bar{r}_\psi(\omega, \upsilon)$ is modeled with a tabular model, which is parameterized by a matrix $\psi$ of shape $|\Omega| \times |\Upsilon|$.

- The learning process is ideal, i.e., consider the PbRL problem as a ranking problem where the true order is defined by the reward function $\bar{r}(\omega, \upsilon)$, and suppose the model can fully fit all preferences provided during training. Such an oracle learning process exists. For instance, the algorithm could utilize a decision forest to comprehensively represent all provided preferences, where each parent node is more preferred than its child nodes.

**Lemma 2.** *Recover the order of $n$ elements needs $C(n) = \log_2(n!) = \mathcal{O}(n \log n)$ queries.*

*Proof.* Consider a set of $n$ arbitrary elements, where the total number of possible sorting outcomes is $n!$. The ranking process can be modeled with a decision tree, where each leaf node corresponds to a distinct sorted outcome, and each branch represents a comparison between two elements. The height of this decision tree reflects the minimum number of comparisons required to arrive at a correctly ordered result, i.e., the number of comparisons required can be expressed as $C(n) = \log_2(n!)$. Then, it is intuitive to check $\log_2(n!) = \mathcal{O}(n \log n)$, which concludes the proof. $\square$

**Proposition 2.** *In the worst case scenario, the conventional PbRL needs $\mathcal{O}(|\Omega||\Upsilon| \log(|\Omega||\Upsilon|))$ additional queries to learn the optimal policy compared to the stage-aligned PbRL.*

*Proof.* We consider the worst-case scenario, where the optimal solution can only be determined after recovering the total order relation of action $\upsilon$ for each stage $\omega$, i.e., $(\omega_i, \upsilon_0), (\omega_i, \upsilon_1), \ldots, (\omega_i, \upsilon_{|\Upsilon_i|})$ is ordered for each $\omega_i$.

Stage-aligned PbRL ranks all actions from each stage, which requires $c_1 = |\Omega| \cdot C(|\Upsilon|) = |\Omega| \log((|\Upsilon|)!)$ queries to learn the optimal policy.

Conventional PbRL ranks all stage-action pairs $(\omega, \upsilon)$. If the $|\Omega|$ groups of actions are ordered, the optimal policy is derived. Similar to Lemma 2, there are $(|\Omega||\Upsilon|)!$ possible choices, and $(|\Omega|)!$ of them are correct. Therefore, conventional PbRL requires $c_2 = \log((|\Omega||\Upsilon|)!) - \log((|\Omega|)!)$ queries to learn the optimal policy.

Compare $c_1$ and $c_2$, we have

$$\exp(c_2 - c_1) = \frac{(|\Omega||\Upsilon|)!}{e^{|\Omega|}(|\Upsilon|)!(|\Omega|)!} \approx \frac{1}{\sqrt{2\pi}} \frac{|\Omega|^{|\Omega|(|\Upsilon|-1)}|\Upsilon|^{|\Upsilon|(|\Omega|-1)}}{e^{|\Omega|+|\Omega||\Upsilon|-(|\Omega|+|\Upsilon|)}}, \tag{20}$$

where the approximation is based on Stirling's formula, i.e., $n! \sim \sqrt{2\pi n} \left(\frac{n}{e}\right)^n$. As $|\Omega|, |\Upsilon| \gg e$, we simplify the above equation to

$$\exp(c_2 - c_1) \approx \frac{1}{\sqrt{2\pi}} \frac{|\Omega|^{|\Omega||\Upsilon|}|\Upsilon|^{|\Omega||\Upsilon|}}{e^{|\Omega||\Upsilon|}} = \frac{1}{\sqrt{2\pi}} \left(\frac{|\Omega||\Upsilon|}{e}\right)^{|\Omega||\Upsilon|} \gg 1. \tag{21}$$

Then, we have

$$c_2 - c_1 = \mathcal{O}(|\Omega||\Upsilon| \log(|\Omega||\Upsilon|)), \tag{22}$$

Which concludes the proof. $\square$

**Proposition 3.** *If the stage reward bias is sufficiently large, such that the reward ordering between $(\omega_i, \upsilon_j)$ and $(\omega_{i'}, \upsilon_{j'})$ depends solely on $\omega_i$ and $\omega_{i'}$ for $i \neq i'$, then in the worst case scenario, conventional PbRL requires $\mathcal{O}(|\Omega|^2|\Upsilon| \log(|\Upsilon|))$ additional queries to learn the optimal policy compared to stage-aligned PbRL.*

*Proof.* In scenarios with large stage bias, queries with $i \neq i'$ do not contribute to the ranking (in fact, only $|\Omega|$ of them do some contribution, as to determine the order of $\omega_i$).

The probability to sample $\omega_i, \omega_{i'}$ s.t. $i = i'$ in all stage-action pair is

$$\frac{|\Omega| \cdot C_{|\Upsilon|}^2}{C_{|\Omega||\Upsilon|}^2} = \frac{|\Omega||\Upsilon|(|\Upsilon|-1)}{|\Omega||\Upsilon|(|\Omega||\Upsilon|-1)} \approx \frac{1}{|\Omega|}, \tag{23}$$

As stage-aligned PbRL ranks all actions from each stage, which requires $c_1 = |\Omega| \cdot C(|\Upsilon|) = |\Omega| \log((|\Upsilon|)!)$ queries to learn the optimal policy, the conventional PbRL requires $\mathcal{O}(|\Omega|^2|\Upsilon| \log(|\Upsilon|))$ queries to derive the optimal policy. $\square$

## A.3 Theoretical Analysis for the Quadrilateral Distance

**Proposition 5.** *Let $\Omega$ be the set of stages and $d_{SD} : \mathcal{S} \times \mathcal{S} \to \mathbb{R}^+$ be a quasimetric temporal distance function. Assume:*

*(1) (Ideal stage partition) For each state $s \in \mathcal{S}$, there exists a unique stage $\omega \in \Omega$ such that $F(s, \omega) = 1$, denoted as $s \in \omega$.*
*(2) (Stage compactness) For any stage $\omega \in \Omega$, there exists $\delta_s^+ \geq 0$ s.t. $\forall s, s' \in \omega, d_{SD}(s, s') \leq \delta_s^+$.*
*(3) (Cross-stage separation) There exists $\delta_c^- > 0$ s.t. if $s \in \omega$ and $s' \in \omega'$ with $\omega \neq \omega'$, then $d_{SD}(s, s') \geq \delta_c^-$.*
*(4) (Boundedness) There exists $\Delta^+ \geq \delta_s^+$ s.t. $\forall s, s' \in \mathcal{S}, d_{SD}(s, s') \leq \Delta^+$.*

*For two segments $\sigma_0 = (s_0^0, s_0^{H-1})$ and $\sigma_1 = (s_1^0, s_1^{H-1})$, we define the quadrilateral distance $d_{stage}(\sigma_0, \sigma_1)$ as in (7). As visualized in Figure 4, $d_{stage}(\sigma_0, \sigma_1)$ is calculated as the average value of temporal distances between the two start and end points of the two segments:*

$$d_{stage}(\sigma_0, \sigma_1) = \frac{1}{4} \left[ d_{SD}(s_0^0, s_1^0) + d_{SD}(s_0^{H-1}, s_1^{H-1}) + d_{SD}(s_0^0, s_1^{H-1}) + d_{SD}(s_0^{H-1}, s_1^0) \right].$$

*Then, $d_{stage}$ satisfies the upper and lower bounds as in Table 3, and the upper bounds and lower bounds are strictly increasing across each alignment case, indicating smaller distance values for stage-aligned queries.*

Table 3: The upper and lower bounds of quadrilateral distance $d_{\text{stage}}(\sigma_0, \sigma_1)$ in alignment cases.

| Case | Stage Alignment Condition | Bounds of $d_{\text{stage}}(\sigma_0, \sigma_1)$ |
|------|---------------------------|--------------------------------------------------|
| A | All start and end points $\in \omega$ | $0 \leq d_{\text{stage}}(\sigma_0, \sigma_1) \leq \delta_s^+$ |
| B | $s_0^0, s_1^0 \in \omega, s_0^{H-1}, s_1^{H-1} \in \omega'$ | $\delta_c^-/2 \leq d_{\text{stage}}(\sigma_0, \sigma_1) \leq (\delta_s^+ + \Delta^+)/2$ |
| C | $s_0^0, s_1^0 \in \omega$, others $\notin \omega$ | $3\delta_c^-/4 \leq d_{\text{stage}}(\sigma_0, \sigma_1) \leq (\delta_s^+ + 3\Delta^+)/4$ |
| D | All start and end points are in distinct stages | $\delta_c^- \leq d_{\text{stage}}(\sigma_0, \sigma_1) \leq \Delta^+$ |

*Proof.* **Case A (Complete Alignment):** If all endpoints belong to the same stage $\omega$, the stage compactness implies all terms in $d_{\text{stage}}(\sigma_0, \sigma_1)$ to be smaller than $\delta_s^+$. Thus,

$$d_{\text{stage}}(\sigma_0, \sigma_1) \leq \frac{1}{4}(4\delta_s^+) = \delta_s^+. \tag{24}$$

Non-negativity of $d_{\text{SD}}$ ensures $d_{\text{stage}} \geq 0$.

**Case B (Endpoint Alignment):** For the aligned start points $s_0^0, s_1^0 \in \omega$ and the aligned end points $s_0^{H-1}, s_1^{H-1} \in \omega'$, the cross-stage terms in $d_{\text{stage}}(\sigma_0, \sigma_1)$ satisfy $d_{\text{SD}}(s_0^0, s_1^{H-1}) \geq \delta_c^-$ and $d_{\text{SD}}(s_0^{H-1}, s_1^0) \geq \delta_c^-$. Thus, the lower bound and upper bound of $d_{\text{stage}}(\sigma_0, \sigma_1)$ are

$$d_{\text{stage}} \geq \frac{1}{4}(0 + 0 + \delta_c^- + \delta_c^-) = \frac{\delta_c^-}{2}, \tag{25}$$

$$d_{\text{stage}} \leq \frac{1}{4}(\delta_s^+ + \delta_s^+ + \Delta^+ + \Delta^+) = \frac{\delta_s^+ + \Delta^+}{2}. \tag{26}$$

**Case C (Partial Alignment):** With only $s_0^0, s_1^0 \in \omega$, three terms in $d_{\text{stage}}(\sigma_0, \sigma_1)$ involve cross-stage pairs. Thus, the lower bound and upper bound of $d_{\text{stage}}(\sigma_0, \sigma_1)$ are

$$d_{\text{stage}} \geq \frac{1}{4}(0 + \delta_c^- + \delta_c^- + \delta_c^-) = \frac{3\delta_c^-}{4}, \tag{27}$$

$$d_{\text{stage}} \leq \frac{1}{4}(\delta_s^+ + \Delta^+ + \Delta^+ + \Delta^+) = \frac{\delta_s^+ + 3\Delta^+}{4}. \tag{28}$$

**Case D (No Alignment):** In this case, all pairs are cross-stage. Thus, the lower bound and upper bound of $d_{\text{stage}}(\sigma_0, \sigma_1)$ are

$$\delta_c^- \leq d_{\text{stage}} \leq \Delta^+. \tag{29}$$

**Priority Ordering:** From $\Delta^+ > \delta_c^-$, we derive:

$$\delta_s^+ < \frac{\delta_s^+ + \Delta^+}{2} < \frac{\delta_s^+ + 3\Delta^+}{4} < \Delta^+, \tag{30}$$

$$0 < \frac{\delta_c^-}{2} < \frac{3\delta_c^-}{4} < \delta_c^-. \tag{31}$$

Thus, strict inequality $A < B < C < D$ holds for both bounds. $\square$

# B Algorithm Implementation

We illustrate the full process of STAIR as in Algorithm 1 and 2.

---

**Algorithm 1** STAIR

---

**Require:** Feedback frequency $K$, number of queries per feedback session $M$, Total feedback $N_{\text{total}}$, temporal distance update frequency $K_{\text{SD}}$
1: Initialize replay buffer $\mathcal{D}, \mathcal{D}^{\text{SD}}$, feedback buffer $\mathcal{D}^\sigma$
2: Initialize the policy $\pi(a|s)$ with unsupervised pretraining [22]
3: **for** each iteration **do**
4:     Rollout with $\pi(a|s)$ and store $(s, a, r, s')$ into $\mathcal{D}, \mathcal{D}^{\text{SD}}$
5:     **if** iteration % $K = 0$ and $|\mathcal{D}^\sigma| < N_{\text{total}}$ **then**
6:         Select $\{(\sigma_0, \sigma_1)\}_{i=1}^M \sim \mathcal{D}$ using stage-aligned query selection (see Section 4.2)
7:         Query the teacher for preference $\{y\}_{i=1}^M$
8:         Store preference $\{(\sigma_0, \sigma_1, y)\}_{i=1}^M$ into $\mathcal{D}^\sigma$
9:     **end if**
10:    Update the reward model $\hat{r}_\psi$ with $\mathcal{D}^\sigma$ using (2)
11:    **if** iteration % $K_{\text{SD}} = 0$ **then**
12:       Update the temporal distance model with $\mathcal{D}^{\text{SD}}$ using (6)
13:       Set $\mathcal{D}^{\text{SD}} \leftarrow \emptyset$
14:    **end if**
15:    Relabel $\mathcal{D}$ with $\hat{r}_\psi$
16:    Update the policy $\pi(a|s)$ using $\mathcal{D}$
17: **end for**

---

**Algorithm 2** STAGE-ALIGNED QUERY SELECTION

---

**Require:** Number of candidate queries $N_c$, number of queries per feedback session $M$
1: Sample $N_c$ segment pairs $\{(\sigma_0^i, \sigma_1^i)\}_{i=1}^{N_c}$
2: Initialize query selection vector of shape $N_c$ with zeros: $\hat{I} = [0, 0, \ldots, 0]$.
3: **for** each segment pair $(\sigma_0^i, \sigma_1^i)$ **do**
4:     Calculate selection score and store it in $\hat{I}$: $\hat{I}(i) \leftarrow I(\sigma_0^i, \sigma_1^i)$
5: **end for**
6: Select $M$ queries with the largest selection score $\hat{I}$

---

# C   Challenges in Stage-Aligned Reward Learning

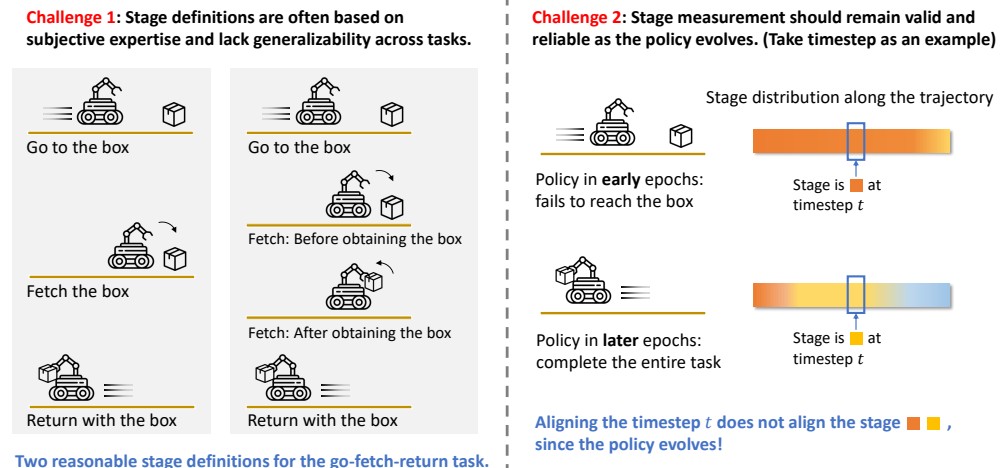

Figure 9: Illustrations of the two challenges that emerged in the design of the stage-aligned reward learning method.

**Stage definitions are often based on subjective expertise and lack generalizability across tasks.** The definition of the stage often requires expert insights and subjective judgment, and can vary significantly across domains and individuals. For instance, in the go-fetch-return example shown in Figure 1, one might opt to further subdivide the retrieval stage into two parts: before and after obtaining the box. Therefore, there is a need for an alternative framework that approximates stages without requiring explicit stage definitions. Meta-learning approaches learning task embeddings for segments [47] or clustering methods [46], could potentially address this challenge. However, these methods are typically complex and demand large amounts of data for effective learning, which is unavailable in PbRL.

**Stage measurement should remain valid and reliable as the policy evolves.**   Take an intuitive approximation, timestep, as an example. For a fixed policy, the timestep serves as a continuous approximation of the stage, as states that are close together on a trajectory tend to belong to the same stage. However, the timestep relies on the sampling policy, which cannot be relabeled once collected. As the policy evolves during the training process and given that state-of-the-art PbRL methods are off-policy for query reuse, segments collected in the early training phases may confuse the query selection in later phases. Therefore, it is essential to develop a stage approximation that is tied to a specific policy and can be easily relabeled according to the trained policy.

# D  Experimental Details

## D.1  Analysis of the Impact of Stage Misalignment

This section provides the setups and implementation details of experiments in Section 3.

**Details on the classification experiment in Section 3.1.**  We conduct the classification experiment in Section 3.1 on Metaworld door-open, drawer-open, and window-open tasks. For each task, we first train a policy using SAC [11] with the ground truth task reward as the expert policy, and then collect 1e6 transitions $(s, r, t)$ with the trained policy, where $t$ is the timestep. The SAC is implemented with the code provided by BPref [21], and is consistent with the SAC used in Section 5. The detailed hyperparameters of SAC are shown in Table 7.

Then, to verify the multi-stage property, we train a classifier which outputs the estimated timestep $\hat{t}$ of the input state $s$. Among the collected $1 \times 10^6$ transitions, we randomly select $5 \times 10^5$ transitions for training the classifier, and report the result that is evaluated on the other $5 \times 10^5$ transitions. The detailed hyperparameters of the classifier are shown in Table 4.

Table 4: Hyperparameters of the timestep classifier.

| Hyperparameter | Value |
|---|---|
| Size of hidden layers | 256 |
| Number of hidden layers | 3 |
| Train epochs | 20 |
| Batch size | 256 |
| Learning rate | $1 \times 10^{-4}$ |
| Optimizer | Adam |

**Details on the human segment preferences experiment in Section 3.2.**  To show the existence of the stage reward bias, where the reward here refers to the underlying reward of humans, we sample segments of length 20 from the SAC trajectories, compose them into queries randomly, and then let humans label preferences. Details about the human experiments are shown in Appendix E.

**Details on the abstract MDP experiment in Section 3.2.**  To show the benefits of stage-aligned query selection in multi-stage problems, we instantiate the abstract MDP introduced in Section 3.2 with $|\Omega| = 101, |\Upsilon_i| = 5, i \in \{1, \cdots, 100, \text{T}\}$, as shown in Figure 2. The reward function is $\bar{r}(\omega, \upsilon) = \bar{r}_{\text{sa}}(\omega, \upsilon) + \bar{r}_{\text{stage}}(\omega)$, where $\bar{r}_{\text{stage}}(\omega)$ denotes the stage reward bias. $\bar{r}_{\text{stage}}(\omega_i) \sim$ Uniform$[0, R_{\text{bias}}], \bar{r}_{\text{sa}}(\omega_i, \upsilon_j) \sim$ Uniform$[0, 10]$. $R_{\text{bias}}$ indicates the strength of the stage reward bias. Additionally, we normalize the reward to ensure a fixed scale across all $R_{\text{bias}}$ values, eliminating the need to tune hyperparameters for policy training. Specifically, we normalize the reward function as follows:

$$\bar{r}'(\omega, \upsilon) = \frac{\bar{r}(\omega, \upsilon) - \frac{1}{|\Omega|} \sum_{\omega'} \min_{\upsilon'} \bar{r}(\omega', \upsilon')}{\frac{1}{|\Omega|} \sum_{\omega'} \max_{\upsilon'} \bar{r}(\omega', \upsilon') - \frac{1}{|\Omega|} \sum_{\omega'} \min_{\upsilon'} \bar{r}(\omega', \upsilon')}, \tag{32}$$

i.e., normalize the reward such that the performance of an arbitrary policy is in $[0, |\Omega|]$. We assume that human preference could be perfectly modeled by this underlying reward function.

In this problem, we use a tabular reward function $\bar{r}_\psi(s, a)$ parameterized by a matrix $\psi$ of shape $|\Omega| \times |\Upsilon|$. The conventional PbRL and the stage-aligned PbRL are implemented as described in Algorithm 3, which details the training process of this reward model based on the Bradley-Terry model. Specifically, lines 3~4 describe the query collection process of the two methods: conventional sampling uniformly samples state-action pairs from the entire state space, while stage-aligned sampling ensures that both state-action pairs come from the same stage. Then, in line 5, the reward model is trained by optimizing the cross-entropy loss.

**Algorithm 3** Tabular PbRL Algorithms for the Abstract MDP

---

**Require:** Number of queries in one epoch $M$.

1: Initialize tabular reward function $\bar{r}_\psi(s, a)$ by setting $\psi$ to a zero matrix.
2: **for** each epoch **do**
3:     *(For conventional PbRL)* Randomly sample $2M$ state-action pairs $\{((s_i, a_i)\}_{i=1}^M$, composing $M$ queries $\mathcal{D} = \{((s_{2i-1}, a_{2i-1}), (s_{2i}, a_{2i}))\}_{i=1}^M$.
4:     *(For stage-aligned PbRL)* Randomly sample $M$ states $\{s_i\}_{i=1}^M$ and $2M$ actions $\{a_i\}_{i=1}^{2M}$, comprising $M$ queries $\mathcal{D} = \{((s_i, a_{2i-1}), (s_i, a_{2i}))\}_{i=1}^M$.
5:     Update $\psi$ by conducting gradient descent with loss function (2) with $\mathcal{D}$.
6: **end for**

---

Hyperparameters of the tabular PbRL algorithms are shown in Table 5.

Table 5: Hyperparameters for policy training in the abstract MDP.

| Hyperparameter | Value |
|---|---|
| Train epochs | 100 |
| Number of queries in one epoch | 200 |
| Learning rate | 0.05 |
| Optimizer | Adam |

### D.2 Tasks

The locomotion tasks from DMControl [40] and robotic manipulation tasks from MetaWorld [44] used in our experiments are shown in Figure 10.

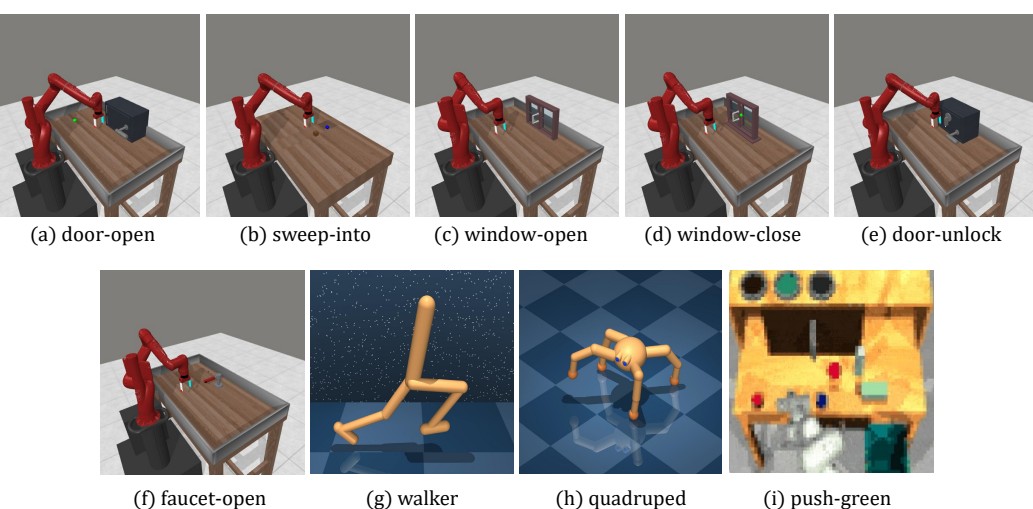

  (a) door-open      (b) sweep-into      (c) window-open      (d) window-close      (e) door-unlock

      (f) faucet-open        (g) walker        (h) quadruped        (i) push-green

Figure 10: Rendered images of tasks from MetaWorld (a-f), DMControl (g-h), and RoboDesk (i).

**Robotic manipulation tasks in MetaWorld.** MetaWorld [44] provides diverse high-dimensional robotic manipulation tasks. We choose the following complex tasks in this work.

- Door-open: An agent controls a robotic arm to open a door with a revolving joint. The door is placed at a random position.
- Sweep-into: An agent controls a robotic arm to sweep a block into the hole. The block is at a random position. The hole is placed at a fixed location.
- Window-open: An agent controls a robotic arm to open a window. The window is placed at a random position.

- Window-close: An agent controls a robotic arm to close a window. The window is placed at a random position.
- Door-unlock: An agent controls a robotic arm to unlock a door by rotating the lock counter-clockwise. The door is placed at a random position.
- Faucet-open: An agent controls a robotic arm to open a faucet by rotating the faucet counter-clockwise. The faucet is placed at a random position.

Please refer to [44] for detailed descriptions of the state space, action space, and the ground truth rewards.

**Locomotion tasks in DMControl suite.** DMControl suite [40] provides diverse high-dimensional locomotion tasks. We choose the following complex tasks in this work.

- Walker-run: A planar walker is trained to control its body and walk on the ground.
- Quadruped-walk: A four-legged ant is trained to control its body and limbs, and crawl slowly on the ground.
- Quadruped-run: A four-legged ant is trained to control its body and limbs, and crawl fast on the ground.

The ground truth reward incorporates components designed to promote forward velocity across all tasks. Additionally, for tasks such as Walker-run, Quadruped-walk, and Cheetah-run, supplementary terms are included to encourage an upright torso posture.

**Robotic manipulation tasks in RoboDesk.** RoboDesk [16] provides diverse robotic manipulation tasks that test diverse behaviors within the same environment. We choose the following complex tasks in this work.

- Push-green: A robotic arm is trained to push the green button to turn the green light on.

We modify the state representation from the original implementation [16], while keeping the action and ground truth rewards unchanged. Specifically, the state is represented as a 68-dimensional vector that combines current and previous timestep information, including the end-effector coordinates (3 dimensions), the wrist joint angle (1 dimension), the gripper finger position (1 dimension), and the positions of all objects in the scenario (29 dimensions).

### D.3 Implementation Details

The implementation of STAIR is based on BPref [21]. Code is available at:

<https://github.com/iiiiiii11/STAIR>

For the implementation of SAC [11], PEBBLE [22], RIME [6], RUNE [23], and QPA [13], we refer to their corresponding official repositories, as shown in Table 6.

Table 6: Source codes of baselines.

| Algorithm | Url | License |
| --- | --- | --- |
| SAC, PEBBLE | <https://github.com/rll-research/BPref> | MIT |
| RUNE | <https://github.com/rll-research/rune> | MIT |
| MRN | <https://github.com/RyanLiu112/MRN> | MIT |
| RIME | <https://github.com/CJReinforce/RIME_ICML2024> | MIT |
| QPA | <https://github.com/huxiao09/QPA> | MIT |

SAC serves as a performance upper bound because it uses the ground-truth reward function, which is unavailable in PbRL settings for training. The detailed hyperparameters of SAC are shown in Table 7. PEBBLE's settings remain consistent with its original implementation, and the specifics are detailed in Table 8. For RUNE, MRN, RIME, QPA and STAIR, most hyperparameters are the same as those

of PEBBLE, and other hyperparameters are detailed in Table 9, 10, 11, 12 and 13, respectively. For RIME, we employ an oracle script teacher in BPref by setting `eps_skip=0`, replacing the noisy teacher, as non-ideal feedback falls outside the scope of this paper. For QPA, we remove the data augmentation to ensure a fair comparison, as none of the other baselines incorporates this technique, and the authors of QPA have noted that data augmentation does not have a significant effect on performance in the MetaWorld environment in the official repository of QPA.[2] The total amount of feedback and feedback amount per session are detailed in Table 14.

The experiments are conducted on a server with Intel(R) Xeon(R) Platinum 8352V CPU, 512 GB RAM, NVIDIA RTX 4090 GPU, and Ubuntu 20.04 LTS. For all baselines and our method, we run 5 different seeds, and report the mean performance and the standard deviation.

Table 7: Hyperparameters of SAC.

| Hyperparameter | Value |
|---|---|
| Number of layers | 2 (DMControl), 3 (MetaWorld) |
| Hidden units per layer | 1024 (DMControl), 256 (MetaWorld) |
| Activation function | ReLU |
| Optimizer | Adam |
| Learning rate | 0.0005 (DMControl), 0.0001 (MetaWorld) |
| Initial temperature | 0.2 |
| Critic target update freq | 2 |
| Critic EMA $\tau$ | 0.01 |
| Batch Size | 1024 (DMControl), 512 (MetaWorld) |
| $(\beta_1, \beta_2)$ | $(0.9, 0.999)$ |
| Discount $\gamma$ | 0.99 |

Table 8: Hyperparameters of PEBBLE.

| Hyperparameter | Value |
|---|---|
| Segment length | 50 |
| Learning rate | 0.0005 (DMControl), 0.0001 (MetaWorld) |
| Feedback frequency | 20000 (DMControl), 5000 (MetaWorld) |
| Num of reward ensembles | 3 |
| Reward model activator | tanh |
| Unsupervised pretraining steps | 9000 |

Table 9: Hyperparameters of RUNE.

| Hyperparameter | Value |
|---|---|
| Initial weight of intrinsic reward $\beta_0$ | 0.05 |
| Decay rate $\rho$ | 0.001 |

---

[2] https://github.com/huxiao09/QPA/issues/1

Table 10: Hyperparameters of MRN.

| Hyperparameter | Value |
|---|---|
| Bi-level updating frequency $N$ | 5000 (Cheetah, Hammer, Button Press), 1000 (Walker) 3000 (Quadruped), 10000 (Sweep Into) |

Table 11: Hyperparameters of RIME.

| Hyperparameter | Value |
|---|---|
| Coefficient $\alpha$ in the lower bound $\tau_{\text{lower}}$ | 0.5 |
| Minimum weight $\beta_{\min}$ | 1 |
| Maximum weight $\beta_{\max}$ | 3 |
| Decay rate $k$ | 1/30 (DMControl), 1/300 (MetaWorld) |
| Upper bound $\tau_{\text{upper}}$ | $3\ln(10)$ |
| $\delta$ for the intrinsic reward | $1 \times 10^{-8}$ |
| Steps of unsupervised pre-training | 9000 |

Table 12: Hyperparameters of QPA

| Hyperparameter | Value |
|---|---|
| Learning rate | 0.0005 (walker-run), 0.0001 (quadruped-walk, quadruped-run, MetaWorld) |
| Size of policy-aligned buffer $N$ | 30 (door-unlock), 60 (door-open), 10 (Other tasks) |
| Data augmentation ratio $\tau$ | 20 |
| Hybrid experience replay sample ratio $\omega$ | 0.5 |
| Min/Max length of subsampled snippets | [35, 45] |

Table 13: Hyperparameters of STAIR.

| Hyperparameter | Value |
|---|---|
| State coefficient of the quadrilateral distance $c_{\text{state}}$ | 0.1 |
| Stage coefficient of the quadrilateral distance $c_{\text{stage}}$ | 2 |
| The frequency of temporal distance update $K_{\text{SD}}$ | 1 |
| Learning rate of the temporal distance | $3 \times 10^{-4}$ |
| Number of layers of Temporal distance | 3 |
| Hidden units per layer | 256 |
| Energy function in Temporal distance | MRN-POT [15] |
| Contrastive loss function in Temporal distance | InfoNCE Symmetric [15] |

Table 14: Feedback amount in each environment. The "value" column refers to the feedback amount in total / per session.

| Environment | Value | Environment | Value |
|---|---|---|---|
| walker-run | 1000/100 | window-open | 2000/50 |
| quadruped-walk | 2000/200 | window-close | 2000/50 |
| quadruped-run | 2000/200 | door-unlock | 5000/50 |
| door-open | 5000/50 | faucet-open | 3000/50 |
| - | 10000/50 | push-green | 1000/50 |
| sweep-into | 10000/50 | | |
| - | 20000/50 | | |

# E  Human Experiments

## E.1  Preference collection

In human experiments, we collect feedback from human labelers (the authors) familiar with the tasks.

**Task 1.**   For the human experiment in Section 3.1, the labelers provide 20 pieces of feedback to each task. Since the task is completed in 100 steps, we only sample segments that begin before the 100th step. Each segment is about 1 second long, which has 20 timesteps. The labelers are instructed to watch a video rendering each segment and determine which one performs better in achieving the specified objective. For each query, the labelers are presented with three options: (1) $\sigma_0$ is better, (2) $\sigma_1$ is better, and (3) the two segments are indistinguishable.

**Task 2.**   For the human experiment in Section 5.3, the labelers provide 50 pieces of feedback to each task. Each segment is about 2 seconds long, which has 50 timesteps. To ensure fairness, we shuffle the queries generated by the algorithms so that the labelers do not know the algorithm that generates the queries. The labelers are instructed to watch a video rendering each segment and determine whether the two segments are in the same stage of this task from their perspective. For each query, the labelers are presented with two options: (1) the two segments are in the same stage, (2) the two segments are in different stages.

## E.2  Guidance to human labelers

Below, we provide the instructions we provided to the human labelers. The instructions are inspired by [7, 18, 29].

**Door-open.**   The target behavior is that the robot arm smoothly rotates the door until it stays fully open at a clearly visible angle. (For task 1 only) If the arm moves abnormally, lower your priority for the segment.

**Drawer-open.**   The target behavior is that the drawer is fully extended to its final position with controlled, direct pushing. (For task 1 only) If the arm moves abnormally, lower your priority for the segment.

**Window-open.**   The target behavior is that the window slides horizontally to a clearly open position with coordinated gripper guidance. (For task 1 only) If the arm moves abnormally, lower your priority for the segment.

**Window-close.**   The target behavior is that the window slides horizontally to a clearly closed position with coordinated gripper guidance. (For task 1 only) If the arm moves abnormally, lower your priority for the segment.

# F   More Experimental Results

## F.1   Multi-Stage Property and Stage Misalignment

In this section, we provide additional results for Section 3. Figure 11 provides additional human experiments validating the existence of stage reward bias in MetaWorld domains, which serves as an example for practical tasks. Figure 12 provides the episode reward and reward estimation error of different $R_{\text{bias}}$ in the abstract MDP model.

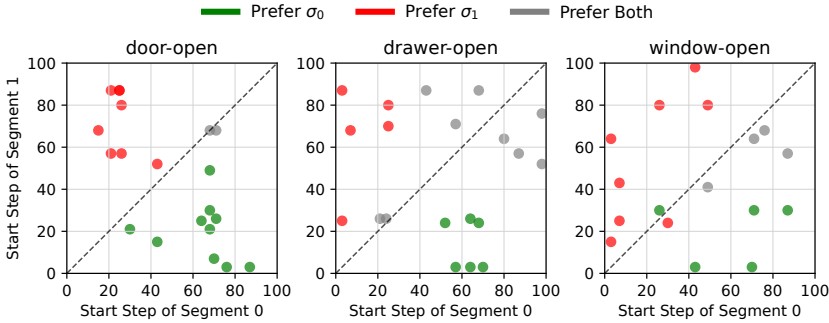

Figure 11: Human Preferences of segments started at different timesteps in the MetaWorld. Each point $(t_x, t_y)$ represents that segment 0 and segment 1 are collected from steps $t_x$ and $t_y$ respectively. Humans prefer segments in later timesteps, suggesting a stage reward bias where the humans' underlying reward is higher in these later stages.

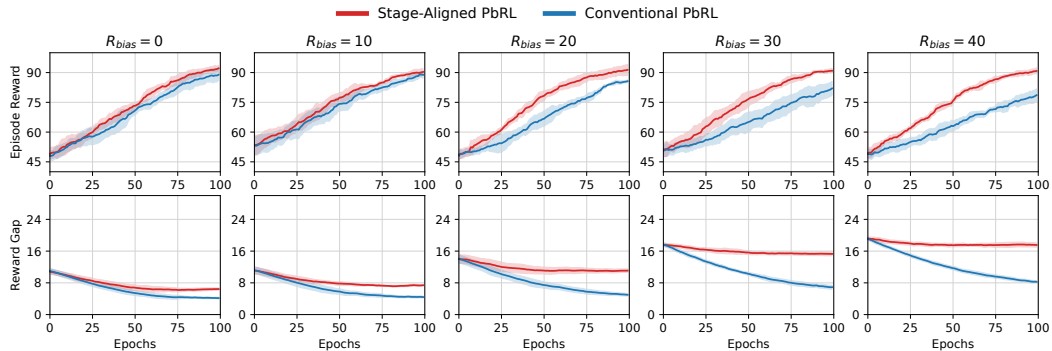

Figure 12: Episode reward (upper row) and reward estimation error (lower row) of different $R_{\text{bias}}$ in the abstract MDP model. The solid line and the shaded area represent the mean and the standard deviation of the corresponding value.

## F.2   Query Visualization

In Figure 13, we visualize the queries selected by PEBBLE and STAIR, respectively. The segment pair $(\sigma_0, \sigma_1)$ selected by PEBBLE is in different stages: the window is already closed in $\sigma_0$, while the arm is in the process of closing the window in $\sigma_1$. Comparing these two segments does not provide the arm with sufficient information to learn how to perform the task of closing the window. In contrast, for the segment pair $(\sigma_0', \sigma_1')$ selected by STAIR, the arm is closing the window in two different ways, and the two segments are within the same stage. This comparison directly provides the arm with actionable information on the mechanics of closing the window, making it more beneficial for learning.

## F.3   Performance on noisy feedback

To evaluate STAIR's robustness to noisy feedback, which mimics imperfect and inconsistent human feedback, we conduct experiments considering two types of "scripted teachers", following prior work

Query selected by PEBBLE: $(\sigma_0, \sigma_1)$

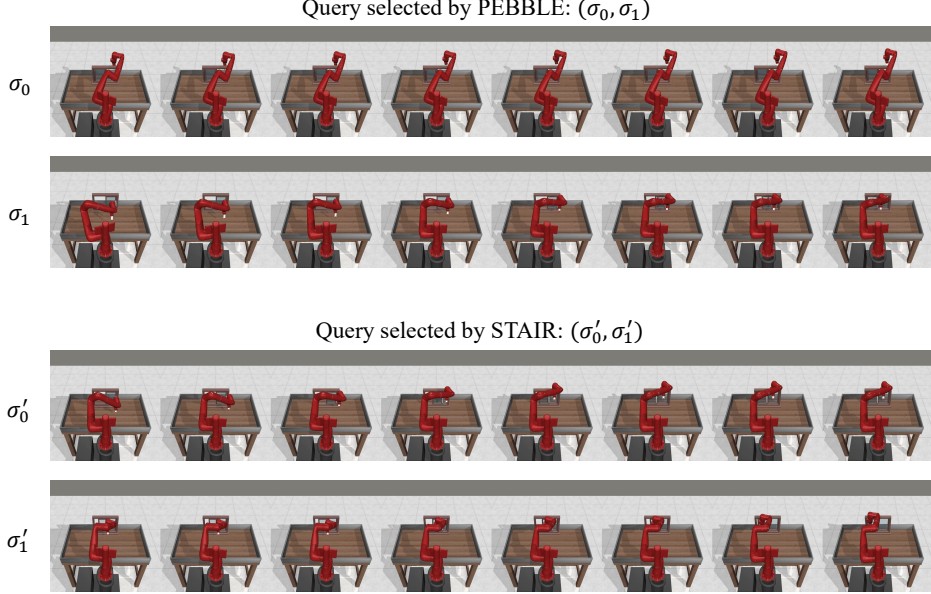

Query selected by STAIR: $(\sigma_0', \sigma_1')$

Figure 13: Visualization of segment pairs selected by (a) PEBBLE (disagreement-based query selection) and (b) STAIR (stage-aligned query selection), under the window-open task.

[21]. Specifically, we consider two types of "scripted teachers": (1) Error teacher: A teacher with a random error rate $\epsilon = 0.1$, resulting in 10% incorrect feedback. (2) Inconsistent teacher: Feedback is randomly sampled from a mixture of two sources: a myopic teacher with discounted factor $\gamma = 0.9$, and an error teacher with $\epsilon = 0.2$.

We show the results on door-open ($N_{\text{total}} = 5000$) and sweep-into ($N_{\text{total}} = 10000$) in Table 15. As shown in the table, STAIR consistently outperforms baselines under both conditions, highlighting its robustness to non-ideal feedback.

Table 15: Performance on noisy feedback.

| Teacher | Door-open | | Sweep-into | |
|---|---|---|---|---|
| | Error | Inconsistent | Error | Inconsistent |
| STAIR | $99.89 _{\pm\ 0.09}$ | $98.53 _{\pm\ 1.36}$ | $49.12 _{\pm 14.71}$ | $56.67 _{\pm 11.18}$ |
| PEBBLE | $91.41 _{\pm\ 6.61}$ | $88.83 _{\pm\ 7.29}$ | $29.64 _{\pm 11.66}$ | $29.86 _{\pm 14.41}$ |
| RUNE | $63.15 _{\pm 13.50}$ | $75.97 _{\pm 13.28}$ | $11.82 _{\pm\ 5.88}$ | $10.66 _{\pm\ 8.76}$ |

# G    Further Discussions

**Explanation for performance in single-stage tasks.**    In single-stage tasks, the performance of STAIR primarily comes from the induced implicit curriculum learning mechanism, where the method adaptively adjusts the learning focus based on the evolving policy.

To explain how the curriculum learning works, we use the Quadruped task as an example. In the quadruped task, early training with STAIR might prioritize selecting segments before and after a fall (which has a small temporal distance), helping the agent learn stability. As training progresses and the policy improves (with the quadruped becoming more stable), the temporal distance between such segments (before and after a fall) increases. At this point, STAIR shifts its focus to segments where the quadruped shows different movement behaviors, rather than emphasizing stability-related segments. This gradual shift enables the agent to learn better movement behaviors while avoiding excessive focus on already-learned behaviors like maintaining stability.

This induced automatic curriculum learning mechanism implicitly divides the reward learning process into stages by introducing queries progressively. In this way, later learning stages (e.g., learning how to walk faster) are presented only after the agent masters the earlier ones (e.g., ensuring stability), enabling the model to focus on the complexities of the newly added stages. Recent works have demonstrated the effectiveness of automatic curriculum learning, which guides the agent with tasks that align with its current capabilities [10, 36].

