# OpenReview forum: "STAIR: Addressing Stage Misalignment through Temporal-Aligned Preference Reinforcement Learning"
_NeurIPS.cc/2025/Conference — NeurIPS 2025 poster_

### Official Review · Reviewer_HXsa · 2025-06-27

**Clarity:** 2
**Significance:** 3
**Originality:** 4
**Rating:** 4
**Confidence:** 2

**Summary:**

The paper addresses a critical challenge in preference-based reinforcement learning (PbRL): stage misalignment in multi-stage tasks. PbRL learns reward functions from human preferences, but when applied to sequential tasks with distinct sub-tasks, comparing segments from different stages leads to uninformative feedback and hinders policy learning. The authors first validate the negative effects of stage misalignment through theoretical analysis and empirical experiments, demonstrating that humans tend to prefer segments from later stages (stage reward bias), and that conventional PbRL methods require significantly more feedback to converge compared to stage-aligned approaches. The authors then propose STAIR, which leverages contrastive learning to approximate stages via temporal distance, and introduces a novel quadrilateral distance for selecting stage-aligned segment comparisons. Extensive experiments on robotic manipulation and locomotion benchmarks show that STAIR outperforms existing PbRL methods in multi-stage tasks and remains competitive in single-stage settings. Human studies further confirm that STAIR's stage approximations align with human cognition, and that STAIR selects stage-aligned queries more effectively than baseline methods.

**Questions:**

1. **Novelty of technical contributions:** Could the authors more clearly distinguish their technical contributions from existing methods, particularly regarding the adaptation of successor distance and contrastive learning to segment distances?
2. **Scope of human experiments:** Would the results hold if STAIR were compared to other state-of-the-art PbRL methods beyond PEBBLE? Could the authors provide statistical significance or effect size measures for the human preference results?
3. **Code and reproducibility:** Could the authors update their code to use current, maintained libraries to ensure long-term reproducibility and adoption?

**Ethical Concerns:**

["NO or VERY MINOR ethics concerns only"]

**Final Justification:**

I thank the authors for their response which addressed my questions, specifically regarding the issue with experimental comparison with more state-of-the-art PbRL methods. After reviewing all the responses, I stand by my positive evaluation of the work overall.

**Limitations:**

The authors briefly mention that the quadrilateral distance only assesses pairwise segment differences, limiting its applicability to other preference formats. The discussion of limitations is quite brief and could be expanded.

**Paper Formatting Concerns:**

No formatting issues.

**Quality:**

4

**Strengths And Weaknesses:**

STRENGTHS:

* **Quantification of stage misalignment:** The paper provides clear empirical and theoretical evidence for the negative impact of stage misalignment in PbRL.
* **Human preference analysis:** The inclusion of human experiments demonstrates that humans prefer segments from later stages, validating the existence of stage reward bias and supporting the motivation for stage-aligned methods.
* **Novel algorithm:** STAIR adapts successor distance and contrastive learning to segment distance, and introduces a new quadrilateral distance for temporal alignment. This combination demonstrates improved performance over existing methods.
* **Performance and generalisation:** STAIR outperforms previous models in multi-stage tasks and shows faster convergence even in single-stage settings, indicating robust generalisation.
* **Human-aligned query selection:** Human studies show that STAIR selects stage-aligned queries more effectively than PEBBLE, suggesting that its stage approximation aligns well with human cognition.

WEAKNESSES:

* **Novelty concerns:** STAIR's adaptation of successor distance and contrastive learning to segment distance raises questions about the novelty of its technical contributions, as these techniques have been introduced in prior work.
* **Limited human experiment comparisons:** Human experiments only compare STAIR to PEBBLE and do not include other state-of-the-art PbRL methods. The statistical significance of the results is not explicitly reported or quantified.
* **Code dependency:** The code relies on deprecated packages (such as old versions of Mujoco), which may limit reproducibility and long-term adoption.

---

> ### Author Rebuttal · Authors · 2025-07-30
>
> Dear Reviewer,
>
> Thanks for your valuable and detailed comments. We hope the following statement clears your concern.
>
>
> **W1, Q1: Novelty concerns.**
>
> **A for W1, Q1:**
> While the successor distance is an established concept, our work introduces significant novelty in the problem setting, motivation, and the proposed method, distinguishing it from prior approaches.
> - **Problem Setting**: Our core contribution lies in adapting the successor distance to estimate stage differences and solve the critical problem of stage misalignment in PbRL, a problem not previously addressed.
> - **Motivation**: Our use of successor distance is well-motivated by the challenges outlined in Appendix C, particularly the need for a dynamic, policy-adaptive stage approximation that avoids task-specific assumptions.
> - **Method**: While the successor distance is defined between individual states, it cannot be directly applied to query selection, which requires a measure between segment pairs. To address it, we tailored the successor distance to our scenario. Specifically, we propose the quadrilateral distance, which extends successor distance to capture temporal and behavioral alignment between segments. This ensures stable and efficient stage-aligned learning in our framework, going beyond the original formulation of successor distance.
>
>
> **W2, Q2: Human experiments lack SOTA baselines and statistical significance analysis.**
>
> **A for W2, Q2:**
> To address your concern, we conduct additional experiments with more state-of-the-art PbRL methods (MRN and RUNE) as baselines, and report the statistical significance analysis result.
> Since evaluating these new baselines requires comparisons with STAIR, we updated the original results to ensure consistent human preference assessments across all methods.
>
> As suggested, we perform statistical significance analyses on the results, including confidence interval calculations and independent two-sample t-tests, to confirm the significance of differences between STAIR and baselines.
> - Confidence Intervals.
>   We report the 95\% confidence interval in Table 1. The narrow intervals confirm that STAIR’s performance in learning human-recognized stages is consistent and tightly clustered around the mean.
> - Two-Sample t-Tests.
>   We conduct independent two-sample t-tests for each task, showing that STAIR achieves statistically significant improvements (p<0.05) in learning human-recognized stages.
>
> |Task|Method|Success Rate|95\% CI|t-value|p-value|Significance|
> |:-:|:-:|:-:|:-:|:-:|:-:|:-:|
> |door-open|STAIR|0.78±0.11|0.78±0.09|-|-|-|
> ||PEBBLE|0.51±0.16|0.51±0.14|3.93|<0.01|✔|
> ||MRN|0.5±0.07|0.5±0.06|6.08|<0.01|✔|
> ||RUNE|0.55±0.13|0.55±0.11|3.8|<0.01|✔|
> |window-close|STAIR|0.73±0.14|0.73±0.12|-|-|-|
> ||PEBBLE|0.55±0.2|0.55±0.17|1.97|0.03|✔|
> ||MRN|0.45±0.09|0.45±0.08|4.54|<0.01|✔|
> ||RUNE|0.38±0.17|0.38±0.14|4.34|<0.01|✔|
> |window-open|STAIR|0.72±0.19|0.72±0.16|-|-|-|
> ||PEBBLE|0.40±0.14|0.40±0.12|3.72|<0.01|✔|
> ||MRN|0.53±0.1|0.53±0.09|2.36|0.02|✔|
> ||RUNE|0.48±0.16|0.48±0.14|2.6|0.01|✔|
>
>
> **W3, Q3: Code dependency.**
>
> **A for W3, Q3:**
> We appreciate the reviewer's concern regarding the reproducibility and long-term usability of our code.
> To ensure fair comparisons, our current environment setup aligns with state-of-the-art methods [1-3], and we have provided detailed environment configurations in the code repository to support reproducibility.
> We are committed to improving the usability of our code and will consider updating it to use current, maintained libraries in future revisions or after acceptance to better support long-term adoption.
>
>
> **L1: Detailed discussion of limitations.**
>
> **A for L1:**
> The quadrilateral distance is currently defined to evaluate pairwise segment differences, which are then used to compute scores for segment pairs.
> Therefore, more general feedback formats, such as listwise or scalar feedback, which contain multiple segments in a single query, are not directly supported.
> However, STAIR can be extended to handle these feedback types by converting them into pairwise preferences:
> - Listwise or Ordinal Feedback: When humans rank a set of samples, pairs can be sampled from the ranked list to construct pairwise feedback.
> - Scalar Scores: When humans provide scores for samples, pairs can be generated by comparing the scores to create pairwise preferences.
>
> Extending STAIR to explicitly handle more preference formats is a promising direction for future research. We will consider that in future work.
>
> **References:**
>
> [1] Lee et al. Pebble: Feedback-efficient interactive reinforcement learning via relabeling experience and unsupervised pre-training.
>
> [2] Cheng et al. RIME: Robust Preference-based Reinforcement Learning with Noisy Preferences.
>
> [3] Liang et al. Reward uncertainty for exploration in preference-based reinforcement learning.

---

> > ### Comment · Reviewer_HXsa · 2025-08-04
> >
> > I thank the authors for their response, specifically regarding the experimental comparison with more state-of-the-art PbRL methods. After reviewing all the responses, I stand by my positive evaluation of the work overall.

---

> ### Author Response · Authors · 2025-08-05
> **Thanks for the positive evaluation**
>
> We would like to thank the reviewer for standing by a positive evaluation. We also appreciate the valuable comments, which helped us significantly improve the paper's strengths.

---

### Official Review · Reviewer_AkSW · 2025-07-03

**Clarity:** 3
**Significance:** 3
**Originality:** 3
**Rating:** 4
**Confidence:** 3

**Summary:**

This paper proposes STAIR (STage-AlIgned Reward learning), a method designed to address the "stage misalignment" issue in preference-based reinforcement learning (PbRL) for multi-stage tasks. Traditional PbRL methods often compare trajectory segments from different subtasks (e.g., navigation vs. grasping), resulting in ambiguous human feedback and reduced policy learning efficiency. STAIR learns a temporal distance via contrastive learning to approximate implicit task stages and prioritizes feedback collection from stage-aligned segment pairs. The authors provide theoretical analysis demonstrating how misaligned queries increase sample complexity and show, through both synthetic and real-world robotic manipulation tasks, that STAIR improves learning efficiency and feedback utilization. Human experiments further validate that STAIR's stage approximation aligns with human cognitive perception.

**Questions:**

1.	Is the temporal distance learning process sensitive to environment reversibility? Can it generalize to irreversible domains or single-shot decision tasks?


2.    Does STAIR still work under noisy or inconsistent manual feedback (e.g., label flipping, annotator disagreement)? Please give a reasonable explanation or more experimental demonstration.

**Ethical Concerns:**

["NO or VERY MINOR ethics concerns only"]

**Limitations:**

1.	The current method only supports binary preference labels, while in real-world settings, human feedback is often more nuanced (e.g., graded or ranked).

2.	The method heavily depends on the quality of contrastively learned temporal embeddings, yet this step lacks ground truth supervision or calibration.

3.	The need to compute quadrilateral distances for many segment pairs during inference could become a bottleneck in large-scale or real-time applications.

**Paper Formatting Concerns:**

There are no formatting issues.

**Quality:**

3

**Strengths And Weaknesses:**

Strengths

1.	The paper clearly identifies stage misalignment as a fundamental but underexplored challenge in PbRL for multi-stage tasks.

2.	The authors rigorously derive complexity gaps between stage-aligned and conventional query selection through abstract MDP constructions (Propositions 2 & 3).

3.	The use of contrastive learning for temporal distance estimation and the novel quadrilateral distance for measuring segment differences are elegant and principled.

4.	The evaluation spans both multi-stage (MetaWorld, RoboDesk) and single-stage (DMControl) tasks, demonstrating robustness, generalizability, and superior performance.

5.	STAIR does not rely on task-specific stage definitions or priors, making it easy to integrate into existing PbRL frameworks.

Weaknesses

1.	While STAIR shows advantages in single-stage tasks, it lacks a deeper theoretical explanation for why stage alignment is still beneficial when explicit stages are absent.

2.	STAIR focuses solely on pairwise preference queries and does not extend to listwise, ordinal, or scalar feedback, which are often more expressive.

3.	Proposition 3 assumes a strong stage reward bias (i.e., human preferences depend primarily on stage position), which may be unrealistic or task-dependent.

4.	Though contrastive learning and quadrilateral distance are tractable, their performance is sensitive to the update frequency (K_SD), which could increase deployment cost in real systems.

---

> ### Author Rebuttal · Authors · 2025-07-30
>
> Dear Reviewer,
>
> Thanks for your valuable and detailed comments. We hope the following statement clears your concern.
>
>
> **W1: Explanation for performance in single-stage tasks.**
>
> **A for W1:**
> In single-stage tasks, the performance of STAIR primarily comes from the induced implicit curriculum learning mechanism, where the method adaptively adjusts the learning focus based on the evolving policy.
>
> To explain how the curriculum learning works, we use the quadruped task as an example.
> In the quadruped task, early training with STAIR might prioritize selecting segments before and after a fall (which has a small temporal distance), helping the agent learn stability.
> As training progresses and the policy improves (with the quadruped becoming more stable), the temporal distance between such segments (before and after a fall) increases.
> At this point, STAIR shifts its focus to segments where the quadruped shows different movement behaviors, rather than emphasizing stability-related segments.
> This gradual shift enables the agent to learn better movement behaviors while avoiding excessive focus on already-learned behaviors like maintaining stability.
>
> This induced automatic curriculum learning mechanism implicitly divides the reward learning process into stages by introducing queries progressively. In this way, later learning stages (e.g., learning how to walk faster) are presented only after the agent masters the earlier ones (e.g., ensuring stability), enabling the model to focus on the complexities of the newly added stages.
> Recent works have demonstrated the effectiveness of automatic curriculum learning, which guides the agent with tasks that align with its current capabilities [1-2].
>
> We will explore a theoretical explanation of the connection between curriculum learning and STAIR in future work.
>
>
>
> **W2, L1: Extension to general feedback formats.**
>
> **A for W2, L1:**
> While we agree that STAIR focuses only on pairwise feedback (as noted in the limitations section), other feedback formats can be converted into binary preferences, enabling potential extension of STAIR:
> - Listwise or ordinal feedback: When humans rank a set of samples based on preference, we can sample pairs from the ranked list to construct binary feedback.
> - Scalar scores: When humans assign scores to samples based on certain rules or subjective opinions, we can sample pairs from the scored dataset and construct binary preferences by comparing the scores.
>
> Moreover, though human feedback in real-world scenarios can often be more expressive, obtaining such detailed feedback is typically challenging [3].
> In contrast, binary preference feedback remains a common approach in practice [4-6].
> Therefore, we prioritize binary preference and leave non-binary preference for future work.
>
>
> **W3: Assumption on stage reward bias.**
>
> **A for W3:**
> While Proposition 3 does assume a strong stage reward bias, we would like to clarify that our method and experiments do not rely on this assumption.
> Instead, Proposition 3 highlights a specific scenario where our algorithm exhibits greater performance advantages.
> Moreover, for more general cases (without the assumption of stage reward bias), Proposition 2 shows that our method remains effective.
>
>
> **W4, L3: Deployment cost due to update frequency ($K_{SD}$).**
>
> **A for W4, L3:**
> We acknowledge that the performance of STAIR requires frequent contrastive learning updates (small $K_{SD}$), which introduces additional computational cost.
> However, despite this frequent update, the overall complexity of STAIR does not increase significantly.
> The primary computational burden of contrastive learning update and quadrilateral distance calculation lies in the temporal distance model, and we analyze its impact during training and inference, respectively:
> - **Training**: STAIR's training time is approximately **1.27 times that of PEBBLE**, which is a manageable increase. This is because the temporal distance model is a three-layer MLP, and its update frequency does not exceed the policy. Additionally, the frequency at which STAIR computes quadrilateral distances for segment pairs is comparable to the frequency at which PEBBLE computes disagreement for segment pairs.
> - **Inference**: During inference, STAIR relies exclusively on the SAC policy, with no additional components applied. Therefore, its inference time matches that of existing methods like PEBBLE or SAC, ensuring no increase in runtime complexity during deployment.
>
>
> **Q1: Sensitivity and extension to environment reversibility.**
>
> **A for Q1:**
> The core mechanism of STAIR is based on the learning of temporal distance [7]. Its applicability depends on whether the temporal distance is well-defined for the given domain. Below, we consider the two specific cases you mentioned:
> - Irreversible tasks: The temporal distance learning algorithm proposed in [7] does not assume environment reversibility. Therefore, STAIR can be extended to irreversible tasks.
> - Single-shot decision tasks: Single-shot decision tasks involve only single-step transitions. Temporal distance and stage concepts cannot be meaningfully defined in these tasks. Consequently, STAIR is not applicable in such scenarios.
>
>
> **Q2: Performance on noisy/inconsistent feedback.**
>
> **A for Q2:**
> As suggested, we conduct experiments to evaluate STAIR's robustness to noisy feedback, which mimics imperfect and inconsistent human feedback. Following prior work [8], we consider two types of "scripted teachers":
> 1. Error teacher: A teacher with a random error rate $\epsilon=0.1$, resulting in 10\% incorrect feedback.
> 2. Inconsistent teacher: Feedback is randomly sampled from a mixture of two sources: a myopic teacher with discounted factor $\gamma=0.9$, and an error teacher with $\epsilon=0.2$.
>
> Results on door-open ($N_\text{total}=5000$) and sweep-into ($N_\text{total}=10000$) show that STAIR consistently outperforms baselines under both conditions, highlighting its robustness to non-ideal feedback.
>
> ||Door-open||Sweep-into||
> |:-:|:-:|:-:|:-:|:-:|
> |Teacher|Error|Inconsistent|Error|Inconsistent|
> |STAIR|**99.89±0.09**|**98.53±1.36**|**49.12±14.71**|**56.67±11.18**|
> |PEBBLE|91.41±6.61|88.83±7.29|29.64±11.66|29.86±14.41|
> |RUNE|63.15±13.50|75.97±13.28|11.82±5.88|10.66±8.76|
>
>
> **L2: Temporal embedding lacks supervision.**
>
> **A for L2:**
> We acknowledge that STAIR depends on the quality of contrastively learned temporal embeddings.
> However, similar to existing methods leveraging temporal distance [7], STAIR only requires the relative ordering of temporal distances between states rather than their exact values, as temporal distance is used to rank segments.
> Therefore, contrastive learning enables effective learning of this relative ordering without requiring ground truth supervision, consistent with prior works adopting similar techniques [9].
>
> We sincerely thank the reviewer again for the timely and valuable comments. We hope that our response has cleared most of your concerns.
>
> **References:**
>
> [1] Florensa, Carlos, et al. Automatic Goal Generation for Reinforcement Learning Agents.
>
> [2] Racaniere, Sebastien, et al. Automated curriculum generation through setter-solver interactions.
>
> [3] Choi et al. Listwise Reward Estimation for Offline Preference-based Reinforcement Learning.
>
> [4] Lee et al. Pebble: Feedback-efficient interactive reinforcement learning via relabeling experience and unsupervised pre-training.
>
> [5] Cheng et al. RIME: Robust Preference-based Reinforcement Learning with Noisy Preferences.
>
> [6] Liang et al. Reward uncertainty for exploration in preference-based reinforcement learning.
>
> [7] Myers, Vivek, et al. Learning temporal distances: Contrastive successor features can provide a metric structure for decision-making.
>
> [8] Lee, Kimin, et al. B-pref: Benchmarking preference-based reinforcement learning.
>
> [9] Jiang, Yuhua, et al. Episodic novelty through temporal distance.

---

> > ### Comment · Reviewer_AkSW · 2025-08-06
> >
> > I appreciate the author's great effort in responding to my questions. The author's response addressed all my concerns. I have decided to keep the original positive score.

---

> ### Author Response · Authors · 2025-08-05
> **Looking forward to further discussions!**
>
> Dear reviewer,
>
> We were wondering if our response and revision have cleared all your concerns. In the previous responses, we have tried to address all the points you have raised. In the remaining days of the rebuttal period, we would appreciate it if you could kindly let us know whether you have any other questions, so that we can still have time to respond and address them. We are looking forward to discussions that can further improve our current manuscript. Thanks!
>
> Best regards,
>
> The Authors

---

> ### Author Response · Authors · 2025-08-07
> **Thanks for the positive score**
>
> We would like to thank the reviewer for standing by a positive score. We also appreciate the valuable comments, which helped us significantly improve the paper's strengths.

---

### Official Review · Reviewer_sEyy · 2025-07-03

**Clarity:** 2
**Significance:** 2
**Originality:** 3
**Rating:** 4
**Confidence:** 4

**Summary:**

This paper presents STAIR, a preference-based reinforcement learning (PbRL) algorithm which leverages temporal distance between trajectory segments to select more informative queries for preference labelling. STAIR was specifically designed for multi-stage environments where an agent has to do multiple subtasks, and the preferences between trajectories across tasks are not very informative. Still, empirical results show that STAIR does not negatively affect the performance of single-stage tasks (such as walker-walk in DMControl).

To estimate the temporal distance of trajectory segments, STAIR leverages a quadrilateral distance based on the successor distance, which is itself learnt through contrastive learning.

Experimental results show that STAIR outperforms many recent PbRL algorithms in multi-stage environments (particularly MetaWorld).

**Questions:**

**Main questions**

_Answering the following questions would help address the weaknesses identified above, and raise the paper rating_.

*Q1*: in Eq 3., why does $G^\tau(i) \le G^\tau(i -1)$? I understand $G^\tau(i -1) \le G^\tau(i)$ (ie. stages must be in order), and $G^\tau(i) -1 \le G^\tau(i)$ (once a stage changes, we cannot go back to the previous one).

*Q2*: Proposition 1 hold independently of $|\Omega|$ ($N_\textrm{stage}$), correct? I believe that is the case based on Appendix A. Either way I would recommend to indicate this in the main text.

*Q3*: I was not able to follow Appendix D.1 and Algorithm 3, was exactly is being trained? It would seem that $r_\textrm{stage}$ and $\bar{r}_\textrm{bias}$ are fixed and sampled independently?

*Q4*: In the Abstract MDP formulation, what does T represent in the expression $i = (0, ..., 100, T)$.

*Q5*:  What is the effect of changing $|\Omega|$ and $|\Upsilon|$?

*Q6*: Does learning with $r_\textrm{bias} = 0$ decay into plain PbRL?

*Q7*: How is $d_\textrm{state}$ computed? Is it simply the variance of the current reward function across all states in the buffer D?

*Q8*: How many experiment repeats were carried out to compute the standard deviations in Figs 5, 6, and 7?

*Q9*: The results in section 5.3 are problematic for two reasons: a) No statistical significance analysis is carried out and the error bars are quite large (therefore it is possible there is actually only a sampling difference between STAIR and PEBBLE), b) the labellers where the authors themselves, which likely have an unconscious (and natural) bias to agree with STAIR on the task segmentation. The easiest fix is to redo the experiment with unrelated participants (ideally enough to be able to carry out a statistical significance test).

*Q10*: An ablation for eq.8  with only $d_\textrm{stage}$ and only $d_\textrm{state}$ would help clarify the contribution of each part of STAIR.

*Q11*: In Table 2, how does STAIR perform with 100 and 50 total feedback? Would it be possible to compare to some of the PbRL methods already included in the main results (many of those improved upon PEBBLE on feedback efficiency).


**Minor questions**


_These questions likely will not affect the review rating, but would help with the paper readability_.

*Q12*: In Sec. 3.2, line 311, I would not use _indicate_ but something like _chosen to_ or _reflect_.

*Q13*: Following [24], $d^\pi_\textrm{SD}$ should be defined as $d^\pi_\textrm{SD} = f_\theta(\mathbf{y}, \mathbf{y}) - f_\theta(x, y)$, otherwise the derivations that follow do not work out. I would additionally mention this is the one-step formulation of the successor distance.

*Q14*: Sec 5.4, line 296 should read _higher_ $K_\mathrm{SD}$ rather than _smaller_.

**Ethical Concerns:**

["NO or VERY MINOR ethics concerns only"]

**Final Justification:**

The authors have mostly address the concerns I raised in my review.

* _W1_: The human evaluation of STAIR stages -> authors redid this experiment with new evaluators (unrelated to the paper), and found that STAIR is significantly better at aligning stages with human judgement than the other baselines.

* _W2_: The theoretical questions about the paper were clarified during the rebuttal.

* _W4_: Use of oracle as human-proxy -> authors added experiments with noisy scripted teachers. No evaluation with actual human feedback was provided (this is the main reason I have not increased my score further).

* _W5_: Missing citation -> This was clarified during rebuttal.

**Limitations:**

Limitations are adequately addressed.

**Paper Formatting Concerns:**

In the appendix please use proper scientific notation for large numbers (eg $5\times10^5$ instead of $5e5$).

**Quality:**

3

**Strengths And Weaknesses:**

**Strengths**

* The research questions are interesting, and address the claims of the paper (namely that STAIR helps with multi-stage tasks, that it does not negatively affect the performance of single-stage task, and that the quadrilateral distance is an adequate form of trajectory clustering).
* STAIR outperforms many recent PbRL algorithms (PEBBLE, RUNE, MRN QPA, and RIME) in MetaWorld.
* Figure 1 clearly conveys the main ideas behind STAIR.
* The theoretical foundations of STAIR are interesting and (mostly) clear (though see questions below).


**Weaknesses**

* _W1_: The human evaluation of alignment between stages of STAIR and human judgement is problematic, calling into question whether the observed benefits from STAIR come from stage clustering, rather than a form of curriculum learning (as the paper hypothesises may be happening for single stage tasks). See questions below for specifics.
* _W2_: Some theoretical aspects of the paper (Figure 3 (right), section 3.2, and $d_\textrm{state}$) need clarification. See questions below.
* _W4_: There is no evaluation of an agent based on actual human feedback, instead of an oracle. Experiments showcasing the robustness of STAIR to human feedback (or at least oracle noise) would increase the significance of the paper.
* _W5_: No citation is provided for "This issue arises when behaviors from different stages, such as navigation and grasping, are presented to humans for comparison. It leads to ambiguous feedback, as labellers struggle to compare behaviours in distinct subtasks, like efficient movement versus precise manipulation", a statement which is key in justifying STAIR.

---

> ### Author Rebuttal · Authors · 2025-07-30
>
> Dear Reviewer,
>
> Thanks for your valuable and detailed comments.
>
> **W4: Performance on noisy feedback.**
>
> **A for W4:**
> As suggested, we conduct experiments to evaluate STAIR's robustness to noisy feedback, which mimics imperfect and inconsistent human feedback. Following prior work [5], we consider two types of "scripted teachers":
> 1. Error teacher: A teacher with a random error rate ε=0.1, resulting in 10\% incorrect feedback.
> 2. Inconsistent teacher: Feedback is randomly sampled from a mixture of two sources: a myopic teacher with discounted factor γ=0.9, and an error teacher with ε=0.2.
>
> Results on door-open ($N_{total}=5000$) and sweep-into ($N_{total}=10000$) show that STAIR consistently outperforms baselines under both conditions, highlighting its robustness to non-ideal feedback.
>
> ||Door-open||Sweep-into||
> |:-:|:-:|:-:|:-:|:-:|
> |Teacher|Error|Inconsistent|Error|Inconsistent|
> |STAIR|99.89±0.09|98.53±1.36|49.12±14.71|56.67±11.18|
> |PEBBLE|91.41±6.61|88.83±7.29|29.64±11.66|29.86±14.41|
> |RUNE|63.15±13.50|75.97±13.28|11.82±5.88|10.66±8.76|
>
> **W5: Ambiguity of stage-aligned queries.**
>
> **A for W5:**
> We thank the reviewer for highlighting this important aspect.
> The ambiguity in human comparisons across different stages can be understood through the lens of Event Segmentation Theory in cognitive sciences [1-2]. This theory suggests that humans naturally perceive continuous actions as segmented into event boundaries. Comparisons that span these boundaries (i.e., comparisons across different stages) significantly increase cognitive load, thereby leading to ambiguous queries.
>
> **W2, Q1, Q2, Q3: Clarification on theory**
>
> - **Q1: Eq.3:**
> We would like to clarify that Eq.3 does not state $G^\tau(i)\le G^\tau(i-1)$.
> Instead, it states $G^\tau(i)-1\le G^\tau(i-1)\le G^\tau(i)$, where the first inequality ensures that the stage changes at most by 1 at each transition, i.e., no stage can be skipped, and the second ensures that stages progress in order without going back.
>
> - **Q2: Proposition 1:**
> Yes, Proposition 1 applies to arbitrary $|\Omega|\le T$, where $T$ is the number of timesteps of the trajectory. We will indicate it in the main text.
>
> - **Q3: Appendix D.1 and Algorithm 3:**
>   - (1) We briefly introduce the outline of Appendix D.1 and Algorithm 3, respectively.
>     - **Appendix D.1** provides the setups and implementation details of experiments in Section 3.  Specifically, it describes the training process of the classifier in Section 3.1, which demonstrates the multi-stage property by predicting the timestep of a given state. It also introduces the source of human-labeled samples in Figure 3 (left) as well as the details of reward function learning in Figure 3 (right). These experiments reveal the existence of stage reward bias in MetaWorld tasks and analyze its impact by comparing conventional sampling with stage-aligned sampling on the abstract MDP.
>     - **Algorithm 3** details the training process of this reward model based on the Bradley-Terry model. Specifically, lines 3-4 describe the query collection process of the two methods: conventional sampling uniformly samples state-action pairs from the entire state space, while stage-aligned sampling ensures that both state-action pairs come from the same stage. Then, in line 5, the reward model is trained by optimizing the cross-entropy loss.
>
>     In the revised paper, we will add some summary texts as stated above in Appendix D.1 and add some high-level explanations near Algorithm 3 to enhance clarity.
>   - (2) Yes, both $r_\textrm{stage}$ and $\bar{r}_\textrm{bias}$ are environment parameters. They are sampled from predefined distributions.
>
> **Q4, Q8: Clarity of the statement.**
> - **Q4:** $T$ in $i=(0,...,100,T)$ denotes terminate, and $w_T$ is a terminated state in the abstract MDP.
> - **Q8:** We conduct 5 experiments (with different seeds) for each result.
>
> **Q5: The effect of changing $|\Omega|$ and $|\Upsilon|$.**
>
> **A for Q5:**
> As suggested, we conduct experiments using different $|\Omega|$ and $|\Upsilon|$ on the abstract MDP, with a fixed stage reward bias of 20, and rewards normalized to be at most 100 across all environments.
> We report the normalized episode reward for each method.
> Results show that stage-aligned PbRL consistently outperforms conventional PbRL.
> Furthermore, as $|\Omega|$ or $|\Upsilon|$ increases, the performance delta in normalized episode reward also increases. This matches Propositions 2 and 3, which state that the additional queries required by conventional PbRL to learn the optimal policy scale with $|\Omega|$ or $|\Upsilon|$.
>
> |$\vert\Omega\vert$|$\vert\Upsilon\vert$|Stage-Aligned PbRL|Conventional PbRL|Delta|
> |:-:|:-:|:-:|:-:|:-:|
> |100|5|90.74±1.89|85.65±1.34|5.09|
> |50|5|94.92±2.44|90.22±4.86|4.69|
> |200|5|83.48±1.02|76.55±0.80|6.93|
> |100|3|95.62±1.69|93.06±1.08|2.55|
> |100|8|82.73±2.47|74.40±2.91|8.32|
>
> **Q6: Does learning with $r_\textrm{bias}=0$ decay into plain PbRL.**
>
> **A for Q6:**
> Learning with $r_\textrm{bias}=0$ does not decay into plain PbRL, as it still selects stage-aligned queries in multi-stage tasks. Theoretically, the $r_\textrm{bias}=0$ case is analyzed in Proposition 2, where our method shows a feedback efficiency advantage over plain PbRL. Experimentally, this is supported by the results in Appendix F.1 (Figure 12), where our method outperforms plain PbRL when $r_\textrm{bias}=0$.
>
> **Q7: How is $d_\textrm{state}$ computed:**
>
> **A for Q7:**
> $d_\textrm{state}(\sigma_0,\sigma_1)$ is computed as the variance of $P_\psi[\sigma_1\succ\sigma_0]$ predicted by an ensemble of reward model.
> Specifically, $d_{\text{state}}(\sigma_0,\sigma_1)=\text{Var}[P_{\psi_i}[\sigma_0\succ\sigma_1]^3_{i=1}]$, where $\{P_{\psi_i}[\sigma_0\succ\sigma_1]\}_{i=1}^3$ are identical reward models with different randomly initalized parameters.
> It is used to identify queries where the ensemble shows higher disagreement, prioritizing more informative queries. This calculation aligns with works in this field, like [3-4].
>
> **W1, Q9: Human experiments.**
>
> **A for W1, Q9:**
> To address your concern, we conducted additional experiments.
>
> (1) Recruitment of participants and mitigation of bias.
>
> As suggested, we invite 8 colleagues from our department to participate in the human experiment.
> To avoid potential bias, queries generated by STAIR and baselines were randomly shuffled, ensuring labelers were unaware of the source algorithm.
> Please note that the original result in section 5.3 is also derived under this mechanism, which ensures fairness, as detailed in Appendix E.1.
>
> (2) Statistical significance analysis.
>
> As suggested, we perform statistical analyses, including confidence interval calculations and independent two-sample t-tests, to confirm the significance of differences between STAIR and baselines.
> - Confidence Intervals.
>   Due to the limited number of samples, the results in Section 5.3 initially had large error bars.
> With additional participants, we collect more feedback, which reduces the standard deviation and narrows the confidence intervals (CIs), confirming STAIR's consistent performance in learning human-recognized stages.
> - Two-Sample t-Tests.
>   We conduct independent two-sample t-tests for each task, showing that STAIR achieves statistically significant improvements (p<0.05) in learning human-recognized stages.
>
> |Task|Method|Success Rate|95\% CI|t-value|p-value|Significance|
> |:-:|:-:|:-:|:-:|:-:|:-:|:-:|
> |door-open|STAIR|0.78±0.11|0.78±0.09|-|-|-|
> ||PEBBLE|0.51±0.16|0.51±0.14|3.93|<0.01|✔|
> ||MRN|0.5±0.07|0.5±0.06|6.08|<0.01|✔|
> ||RUNE|0.55±0.13|0.55±0.11|3.8|<0.01|✔|
> |window-close|STAIR|0.73±0.14|0.73±0.12|-|-|-|
> ||PEBBLE|0.55±0.2|0.55±0.17|1.97|0.03|✔|
> ||MRN|0.45±0.09|0.45±0.08|4.54|<0.01|✔|
> ||RUNE|0.38±0.17|0.38±0.14|4.34|<0.01|✔|
> |window-open|STAIR|0.72±0.19|0.72±0.16|-|-|-|
> ||PEBBLE|0.40±0.14|0.40±0.12|3.72|<0.01|✔|
> ||MRN|0.53±0.1|0.53±0.09|2.36|0.02|✔|
> ||RUNE|0.48±0.16|0.48±0.14|2.6|0.01|✔|
>
> **Q10: Ablation on Eq.8.**
>
> **A for Q10:**
> As suggested, we conduct ablation studies on $d_\textrm{stage}, d_\textrm{state}$ in Eq. 8. The following results show that removing either term degrades performance, demonstrating the effectiveness of the proposed Eq. 8.
>
> ||door-open ($N_{total}=5000$)|sweep-into ($N_{total}=10000$)|
> |:-:|:-:|:-:|
> |STAIR|100.00±0.00|91.08±3.42|
> |STAIR w/o $d_{stage}$|85.57±12.77|59.17±21.62|
> |STAIR w/o $d_{state}$|74.48±15.21|34.86±15.09|
>
> **Q11: Query efficiency experiments.**
>
> **A for Q11:**
> As suggested, we compare STAIR and baselines with 50 and 100 feedback on Metaworld's door-open task. The results show that when $N_{total}\leq 100$, all algorithms fail to perform, likely because such a small amount of feedback is insufficient to learn a reliable reward function.
> However, with $N_{total}=500$, STAIR begins to show strong performance, outperforming all baselines, which shows its effectiveness.
>
> |$N_{total}$|STAIR|PEBBLE|MRN|RUNE|
> |:-:|:-:|:-:|:-:|:-:|
> |50|0±0|0±0|0±0|0±0|
> |100|0±0|0±0|0±0|0±0|
> |500|52.01±23.18|20.00±17.88|19.33±17.29|0.94±0.84|
> |2000|77.77±11.67|28.79±17.02|58.82±21.84|72.03±16.92|
> |5000|100.00±0.00|85.57±12.77|97.40±1.46|72.15±16.45|
> |10000|99.93±0.06|92.53±6.53|93.40±5.46|78.84±17.64|
>
> **Q12-14, PFC: Minor questions, Paper Formatting Concerns:**
> Thank you for your keen attention to detail! We have corrected them in the revised version.
>
> We sincerely thank the reviewer again for the timely and valuable comments. We hope that our response has cleared most of your concerns.
>
> **References:**
>
> [1] Kurby et al. Segmentation in the perception and memory of events.
>
> [2] Zacks et al. Event perception: a mind-brain perspective.
>
> [3] Lee, Kimin et al. Pebble: Feedback-efficient interactive reinforcement learning via relabeling experience and unsupervised pre-training.
>
> [4] Liang, Xinran, et al. Reward uncertainty for exploration in preference-based reinforcement learning.
>
> [5] Lee, Kimin, et al. B-pref: Benchmarking preference-based reinforcement learning.

---

> > ### Comment · Reviewer_sEyy · 2025-08-05
> >
> > Thank you for your detailed rebuttal and the additional experiments and ablations.
> >
> > Following up on some discussion items:
> >
> > **Q3** Apologies for the typo in my review. I was trying to come up with an interpretation for all 3 inequalities in $G^\\tau(i)-1\\le G^\\tau(i-1)\\le G^\\tau(i)$:
> >
> > * $G^\\tau(i)-1\\le G^\\tau(i-1)$ [sequential stages]
> > * $G^\\tau(i-1)\\le G^\\tau(i)$ [not going back]
> > * $G^\\tau(i)-1\\le G^\\tau(i)$ [??].
> >
> > For clarity I would simply add your rebuttal response to the main text.
> >
> > **Q7** This makes sense, as you point out it is how PEBBLE computes the variance. In retrospect, I was confused by the sentence "which calculates the variance of P value across ensemble members (lines 226-227). I think adding a bit more detail and the references you cite would fix this issue.
> >
> > **Q9**: the t-tests are STAIR vs {PEBBLE, MRN, RUNE}, correct? Did you apply the Bonferroni correction for repeated measures in your p-value calculation? (I am assuming you did not redo the STAIR experiment for every comparison with PEBBLE, MRN, and RUNE).
> >
> > Overall, I am satisfied with your rebuttal to all of my other questions, and will correspondingly increase my review score (please allow me some more days to do so though).

---

> > > ### Author Response · Authors · 2025-08-08
> > >
> > > Dear Reviewer sEyy,
> > >
> > > We hope this message finds you well.
> > >
> > > We wanted to express our sincere gratitude for your thorough review and valuable feedback during the rebuttal phase. We particularly appreciated your comment on August 5th, where you indicated your satisfaction with our responses and your intention to increase your review score.
> > >
> > > We understand that you are very busy, but we kindly wanted to check if you had the chance to update your score on the system. We would be very grateful if you could please take a moment to update your official rating on OpenReview to reflect your revised assessment. Please let us know if there's anything else we can clarify or assist with.
> > >
> > > Thank you again for your valuable time and thoughtful review.
> > >
> > > Best regards,
> > >
> > > The Authors

---

> ### Author Response · Authors · 2025-08-05
> **Further response to Reviewer sEyy**
>
> Dear Reviewer,
>
> We greatly appreciate your willingness to improve your score.
> We also appreciate the valuable comments, which helped us significantly improve the paper's strengths.
> We hope the following statement clears your remaining concern.
>
>
> **Q3: Interpretation for Eq.3**
>
> **A for Q3:**
>
> Thank you for pointing out this issue.
> The inequality $G^\tau(i) - 1 \le G^\tau(i)$ is always true (as $-1 \leq 0$), and it indeed holds no specific meaning in this context.
> Our original intention was to represent two key constraints in a more compact form: $G^\tau(i) - 1 \le G^\tau(i-1)$ (ensuring sequential stages) and $G^\tau(i-1) \le G^\tau(i)$ (ensuring no going back).
> To avoid potential confusion, we will present these two inequalities separately in the revised version.
>
> **Q7: Clarity on the calculation of $d_\text{state}$**
>
> **A for Q7:**
>
> Thank you for the helpful feedback. We will revise the manuscript to include additional details and references as suggested.
>
> **Q9: Concerns about data independence and multiple comparisons in human experiments.**
>
> **A for Q9:**
>
> We sincerely thank the reviewer for pointing out this important aspect.
>
> 1. You are correct that we conducted pairwise t-tests between STAIR and each baseline individually (i.e., STAIR vs. PEBBLE, STAIR vs. MRN, and STAIR vs. RUNE). We will explicitly indicate it in the revised version.
>
> 2. Statistical Analysis: We acknowledge that the p-values should indeed be adjusted for multiple comparisons, and we appreciate the reviewer for highlighting this oversight.
>     - To address this concern, we applied the suggested **Bonferroni correction** to the p-values, and the corrected values are provided in the updated table below. After the Bonferroni correction, STAIR remains statistically significant in 7 out of 9 experiments.
>     - To further demonstrate the effectiveness of our method, we also calculated the p-values using the **Benjamini-Hochberg procedure**. As shown in the table, under the Benjamini-Hochberg correction, STAIR achieves statistical significance across all 9 experiments.
>
> These updated results reconfirm that STAIR achieves significant improvements over the baselines in learning human-recognized stages.
> We hope this clarification resolves your concerns.
>
> |Task|Method|Success Rate|95\% CI|t-value|p-value|Bonf-p-value|Bonf-Significance|BH-p-value|BH-Significance|
> |:-:|:-:|:-:|:-:|:-:|:-:|:-:|:-:|:-:|:-:|
> |door-open|STAIR|0.78±0.11|0.78±0.09|-|-|-|-|-|-|
> ||PEBBLE|0.51±0.16|0.51±0.14|3.93|<0.01|<0.01|✔|<0.01|✔|
> ||MRN|0.5±0.07|0.5±0.06|6.08|<0.01|<0.01|✔|<0.01|✔|
> ||RUNE|0.55±0.13|0.55±0.11|3.8|<0.01|<0.01|✔|<0.01|✔|
> |window-close|STAIR|0.73±0.14|0.73±0.12|-|-|-|-|-|-|
> ||PEBBLE|0.55±0.2|0.55±0.17|1.97|0.03|0.11|✖|0.03|✔|
> ||MRN|0.45±0.09|0.45±0.08|4.54|<0.01|<0.01|✔|<0.01|✔|
> ||RUNE|0.38±0.17|0.38±0.14|4.34|<0.01|<0.01|✔|<0.01|✔|
> |window-open|STAIR|0.72±0.19|0.72±0.16|-|-|-|-|-|-|
> ||PEBBLE|0.40±0.14|0.40±0.12|3.72|<0.01|<0.01|✔|0.01|✔|
> ||MRN|0.53±0.1|0.53±0.09|2.36|0.02|0.06|✖|0.02|✔|
> ||RUNE|0.48±0.16|0.48±0.14|2.6|0.01|0.03|✔|0.02|✔|
>
>
> Notes for the table:
> - p-value: The original p-value from the statistical test.
> - Bonf-p-value: The p-value adjusted using the Bonferroni correction.
> - Bonf-Significance: The statistical significance determined based on the Bonferroni-adjusted p-value (Bonf-p-value).
> - BH-p-value: The p-value adjusted using the Benjamini-Hochberg correction.
> - BH-Significance: The statistical significance determined based on the Benjamini-Hochberg-adjusted p-value (BH-p-value).
>
> We sincerely thank the reviewer again for the thoughtful and constructive feedback.
> We hope that our responses and additional experimental results have addressed your concerns.
>
> Best,
>
> The Authors

---

> ### Comment · Reviewer_sEyy · 2025-08-08
>
> Thank you once again for your detailed response, as well as your patience while I considered my review.
>
> In particular, I appreciate the new statistical analysis of the alignment-to-human-stages experiments, it is interesting that not only the results are statistically significant, but also the higher success rate over the baselines.
>
> In light of our discussion, I have decided to increase my score to a 4.

---

### Official Review · Reviewer_iPSk · 2025-07-22

**Clarity:** 3
**Significance:** 3
**Originality:** 3
**Rating:** 4
**Confidence:** 3

**Summary:**

This paper identifies and addresses the problem of stage misalignment in preference-based reinforcement learning (PbRL), where comparisons between behavior segments from different stages of a multi-stage task lead to uninformative or misleading feedback. The authors propose STAIR (STage-AlIgned Reward learning), which learns an on-policy, task-agnostic temporal distance metric via contrastive learning. This metric is used to filter for stage-aligned segment pairs when querying preferences. A novel “quadrilateral distance” measures stage similarity between segments. STAIR significantly improves feedback efficiency and policy performance on multi-stage tasks (MetaWorld, RoboDesk) and remains competitive on single-stage benchmarks (DMControl). A small-scale human study confirms that STAIR-selected queries align well with human perception of stages.

**Questions:**

1. How does STAIR perform with noisy or inconsistent human feedback? Injecting noise into oracle labels could help assess robustness.
2. Can STAIR handle branching or non-linear stage structures? Most experiments assume linear sequential tasks.
3. How could STAIR extend to more general feedback formats (e.g., ranked lists or scalar scores)?

**Ethical Concerns:**

["NO or VERY MINOR ethics concerns only"]

**Final Justification:**

I have read the author response and have no further questions. I'll maintain my score.

**Limitations:**

Yes

**Quality:**

3

**Strengths And Weaknesses:**

Strengths:
1. Strong theoretical justification (Propositions 2–3) and rigorous evaluation.
2. STAIR consistently outperforms PbRL baselines on multi-stage tasks with far fewer queries.
3. The paper is well-written and structured. Key concepts (temporal distance, stage alignment) are clearly explained with useful diagrams.
4. The paper tackles an important underexplored issue in PbRL with implications for real-world learning from humans.
5. The paper enjoys strong potential to improve sample efficiency for long-horizon tasks.

Weaknesses:
1. Human feedback is simulated via an oracle—true robustness to real-world human noise remains untested.
2. The proposed algorithm adds considerable complexity (contrastive learning, extra hyperparameters), which may hinder adoption.
3. The limited demonstration with noisy or real human-in-the-loop training slightly makes real-world application less practical.

---

> ### Author Rebuttal · Authors · 2025-07-25
>
> Dear Reviewer,
>
> Thanks for your valuable and detailed comments. We hope the following statement clears your concern.
>
> **W1, W3, Q1: Robustness on noisy or inconsistent feedback.**
>
> **A for W1, W3, Q1:**
> As suggested, we conduct experiments to evaluate STAIR's robustness to noisy feedback, which mimics imperfect and inconsistent human feedback. Following prior work [1], we consider two types of "scripted teachers":
> 1. Error teacher: A teacher with a random error rate $\epsilon=0.1$, resulting in 10\% incorrect feedback.
> 2. Inconsistent teacher: Feedback is randomly sampled from a mixture of two sources: a myopic teacher with discounted factor $\gamma=0.9$, and an error teacher with $\epsilon=0.2$.
>
> Results on door-open ($N_\text{total}=5000$) and sweep-into ($N_\text{total}=10000$) show that STAIR consistently outperforms baselines under both conditions, highlighting its robustness to non-ideal feedback.
>
> ||Door-open||Sweep-into||
> |:-:|:-:|:-:|:-:|:-:|
> |Teacher|Error|Inconsistent|Error|Inconsistent|
> |STAIR|**99.89±0.09**|**98.53±1.36**|**49.12±14.71**|**56.67±11.18**|
> |PEBBLE|91.41±6.61|88.83±7.29|29.64±11.66|29.86±14.41|
> |RUNE|63.15±13.50|75.97±13.28|11.82±5.88|10.66±8.76|
>
>
> **W2: Complexity concerns.**
>
> **A for W2:**
> To address your concern, we analyze the complexity of contrastive learning and the complexity of the extra hyperparameters, respectively.
>
> (1)
> **As for the complexity**, STAIR introduces additional components, but the overall complexity does not increase significantly.
> We evaluate this in terms of both training and inference:
> - Training: STAIR's training time is approximately 1.27 times that of PEBBLE, which remains manageable. This is because the temporal distance model is a lightweight three-layer MLP, and its update frequency does not exceed the policy.
> - Inference: During inference, STAIR relies exclusively on the SAC policy, with no additional components applied. Therefore, its inference time matches that of existing methods like PEBBLE or SAC, ensuring no increase in runtime complexity during deployment.
>
> (2) **Regarding hyperparameters**, the additional hyperparameters introduced by STAIR are consistent across all environments. Besides, ablation studies in Section 5.4 show the robustness of STAIR to query selection parameters, and provide guidance for the selection of temporal distance update frequency, which can facilitate more effective applications.
>
>
>
> **Q2: Applicability in non-linear stage structures.**
>
> **A for Q2:**
> We believe STAIR can be adapted to handle branching or non-linear stage structures.
> STAIR's core mechanism relies on using temporal distance to measure stage differences.
> The temporal distance used in our work [2], defined as
> $$
> d^{\pi}_{\text{SD}}(x, y) = \log \left( {p^{\pi} _\gamma (s _+ = y | s_0 = y)}/{p^{\pi} _\gamma (s _+ = y | s_0 = x)}\right),
> $$
> which quantifies the transition probabilities between states under a given policy and does not depend on linear or sequential task structures.
> This independence makes it inherently adaptable to more complex scenarios, such as branching or non-linear stage structures, as long as temporal distance appropriately reflects stage differences in these settings.
> Extending STAIR to explicitly handle branching or non-linear tasks, along with analyzing its performance in such contexts, could be a promising direction for future research.
>
>
>
> **Q3: Extension to general feedback formats.**
>
> **A for Q3:**
> STAIR can generalize to broader feedback formats by extracting pairwise feedback from them:
> - Ranked Lists: Given a set of samples ranked by human preference, we can sample pairwise comparisons from the ranked list to construct pairwise feedback.
> - Scalar Scores: When humans assign scalar scores to samples, we can sample pairs from the dataset and use the scores to derive pairwise feedback by comparing their relative magnitudes.
>
> We thank the reviewer for this insightful suggestion, which we will consider in our future work.
>
>
> Thanks again for the valuable comments. We sincerely hope our additional experimental results and explanation have cleared the concern. More comments on further improving the presentation are also very much welcome.
>
> **References:**
>
> [1] Lee, Kimin, et al. B-pref: Benchmarking preference-based reinforcement learning.
>
> [2] Myers, Vivek, et al. Learning temporal distances: Contrastive successor features can provide a metric structure for decision-making.

---

> ### Author Response · Authors · 2025-08-05
> **Looking forward to further discussions!**
>
> Dear reviewer,
>
> We were wondering if our response and revision have cleared all your concerns. In the previous responses, we have tried to address all the points you have raised. In the remaining days of the rebuttal period, we would appreciate it if you could kindly let us know whether you have any other questions, so that we can still have time to respond and address them. We are looking forward to discussions that can further improve our current manuscript. Thanks!
>
> Best regards,
>
> The Authors

---

### Note · Authors · 2025-08-12

Dear Area Chair and Reviewers,

We are sincerely grateful to the Area Chair and all reviewers for their diligent work and thoughtful engagement. Their insightful feedback has been invaluable in refining and strengthening our manuscript.

We are delighted that the reviewers highlighted our work as "novel" (iPSk), addressed a "fundamental but underexplored challenge" (iPSk, AkSW), with "strong theoretical evidence" and "superior performance" (all reviewers).

The primary discussion points were to improve the paper's clarity and further validate its practical applicability.
We have addressed these points thoroughly in our rebuttal and will integrate the following improvements into the revised version:

1. **Robustness to Noisy Feedback**: To demonstrate STAIR's robustness, a key practical concern, we conducted new experiments under simulated noisy conditions ("Error teacher" and "Inconsistent teacher"). Results showed that **STAIR consistently outperformed baselines** (PEBBLE, RUNE), demonstrating its robustness.

2. **Strengthened Human Experiments**: In response to feedback on our human experiments, we expanded the study to include more state-of-the-art baselines (MRN, RUNE) and conducted rigorous statistical tests. Results showed **a statistically significant improvement for STAIR** in aligning with human cognitive stages.

3. **Computational Efficiency**: We thoroughly analyzed STAIR's computational costs for both training and inference, confirming its efficiency with practically manageable complexity.

4. **Generalization and Extensibility**: We clarified STAIR's adaptability by outlining how it can be naturally extended to more general task structures (e.g., branching tasks) and feedback formats (e.g., ranked lists or scalar scores).

5. **Enhanced Clarity of Technical Details**: To improve clarity, we refined the explanations for key equations (Eq. 3), propositions (1 and 3), and experimental settings (Appendix D.1).

We are delighted that our responses were met with such a positive reception, leading to **unanimously positive scores**.
We are particularly grateful to the reviewer who found the "new statistical analysis" compelling and raised the score, and to the others who confirmed that we "addressed all their concerns" and stood by "positive evaluation."

We thank the reviewers once again for the constructive and positive engagement. We also extend our sincere gratitude to the Area Chair for leading the review process.

Best,

The Authors

---

### Decision · Program_Chairs · 2025-09-17

**Decision:**

Accept (poster)

**Comment:**

This work identifies a fundamental and underexplored issue in PbRL, and theoretically demonstrates the gap in multi-stage tasks. It suggests a natural but nontrivial resolution and theoretically demonstrates its benefit as well as
well as perform empirical evaluation which spans both multi-stage (MetaWorld, RoboDesk) and single-stage (DMControl) tasks, demonstrating robustness, generalizability, and superior performance.
Finally, the paper is very well written.
From these reasons the paper should certainly be accepted.